# Southern Ocean phytoplankton under climate change: shifting balance of bottom-up and top-down control

Tianfei Xue[1], Jens Terhaar[2,3,4], A.E. Friederike Prowe[1], Thomas L. Frölicher[3,4], Andreas Oschlies[1], and Ivy Frenger[1]

[1]GEOMAR Helmholtz Centre for Ocean Research Kiel, Kiel, Germany
[2]Department of Marine Chemistry and Geochemistry, Woods Hole Oceanographic Institution, Woods Hole, Massachusetts, USA
[3]Climate and Environmental Physics, Physics Institute, University of Bern, Bern, Switzerland
[4]Oeschger Centre for Climate Change Research, University of Bern, Bern, Switzerland

**Correspondence:** Tianfei Xue (txue@geomar.de)

**Abstract.** Phytoplankton form the base of the marine food web by transforming $CO_2$ into organic carbon via photosynthesis. Despite the importance of phytoplankton for marine ecosystems and global carbon cycling, projections of phytoplankton biomass in response to climate change differ strongly across Earth system models, illustrating uncertainty in our understanding of the underlying processes. Differences are especially large in the Southern Ocean, a region that is notoriously difficult to
represent in models. Here, we argue that total (depth-integrated) phytoplankton biomass in the Southern Ocean is projected to largely remain unchanged under climate change by the CMIP6 multi-model ensemble because of a shifting balance of bottom-up and top-down processes driven by a shoaling mixed layer depth. A shallower mixed layer is projected on average to improve growth conditions, consequently weaken bottom-up control, and confine phytoplankton closer to the surface. An increase of phytoplankton concentration promotes zooplankton grazing efficiency, thus intensifying top-down control. However,
large differences across the model ensemble exist, with some models simulating a decrease in surface phytoplankton concentrations. To reduce uncertainties in projections of surface phytoplankton concentrations, we employ an emergent constraint approach using the observed sensitivity of surface chlorophyll concentration, taken as an observable proxy for phytoplankton, to seasonal changes in the mixed layer depth as an indicator for future changes in surface phytoplankton concentrations. The emergent constraint reduces uncertainties in surface phytoplankton concentration projections by around one third and increases
confidence that surface phytoplankton concentrations will indeed rise due to shoaling mixed layers under global warming, thus favouring intensified top-down control. Overall, our results suggest that while changes in bottom-up conditions stimulate enhanced growth, intensified top-down control opposes an increase in phytoplankton and becomes increasingly important for phytoplankton response to climate change in the Southern Ocean.

## 1 Introduction

Phytoplankton are fundamental to the marine ecosystem and the global carbon cycle, as they are the base of the ocean food web (Pauly and Christensen, 1995) and account for approximately half of the global biological carbon fixation (Field et al., 1998).

Despite the importance of phytoplankton for the carbon cycle and the future of marine ecosystems, the response of phytoplankton to climate change remains uncertain. Earth system model projections of phytoplankton biomass and its growth for the 21st century largely differ between models and scenarios (Steinacher et al., 2010; Laufkötter et al., 2015; Frölicher et al., 2016; Tagliabue et al., 2021) and even the direction of changes remains unclear (Kwiatkowski et al., 2020). The different projections are due to uncertainties in the underlying physical forcing and different biological and biogeochemical parameterizations of processes in the models (Laufkötter et al., 2015). Such differences in the biological and biogeochemical process parameterizations have the potential to propagate through the food web, often amplified by simplistic food web dynamics (Stock et al., 2014b), and cause even greater model differences for higher trophic level projections (Lotze et al., 2019). In turn, such differences in higher trophic level projections and simplistic food web dynamics will, through ill-constrained grazing pressure on phytoplankton, lead to amplified model differences in phytoplankton projections, forming a feedback loop for inter-model differences in simulated ecosystem dynamics. Given the role of phytoplankton in marine ecosystems and biogeochemical cycling, better constraining how phytoplankton respond to changes in their physical, chemical and biological environment under climate change is critical.

Climate change affects phytoplankton through both "bottom-up" (Marinov et al., 2010; Leung et al., 2015) and "top-down" (Stock et al., 2014a; Kwiatkowski et al., 2019) control. Bottom-up control describes changes in phytoplankton growth due to changes in light, temperature, and nutrients caused by environmental factors like mixing or upwelling (Behrenfeld et al., 2006). Given that these environmental factors are projected to change under climate change, it is expected that phytoplankton will also be affected. Top-down control, on the other hand, refers to the impact of predators on phytoplankton populations. The major predator of phytoplankton are zooplankton. Phytoplankton changes are typically considered top-down controlled when zooplankton and phytoplankton changes are inversely related (Chust et al., 2014); for example, when phytoplankton decreases while zooplankton increases, suggesting that zooplankton is grazing down phytoplankton. Globally, changes in bottom-up control are thought to dominate the phytoplankton response to changing climate, with increasing ocean stratification and shoaling mixed layers driving a reduction in nutrient supply and hence a reduction in phytoplankton biomass and growth (Sarmiento et al., 2004b; Bopp et al., 2005; Behrenfeld et al., 2006; Steinacher et al., 2010; Boyce et al., 2010). An opposite bottom-up response can be found in high-latitude regions where improved light conditions due to increased stratification and sea ice retreat are projected to lead to phytoplankton increases (Sarmiento et al., 2004b; Deppeler and Davidson, 2017). Although predators and grazing pressure also evolve with climate change (Shurin et al., 2012), top-down control has received much less attention (Ratnarajah et al., 2023).

The Southern Ocean is a region where models, albeit with an uncertain magnitude, robustly project that phytoplankton will grow better with shoaling mixed layer depths under global warming (Fig. 1 & 2; Sarmiento et al., 2004b; Bopp et al., 2013; Laufkötter et al., 2015; Kwiatkowski et al., 2020). Despite the high abundance of macronutrients (Fig. A1), phytoplankton growth in the Southern Ocean is at present largely bottom-up controlled by iron and light limitations (Martin et al., 1990; Mitchell et al., 1991; Arteaga et al., 2020; Moore et al., 2013). Light limitation is especially strong in the Southern Ocean due to the deep mixed layers and low surface radiation in mid-to-high latitude winters. Under global warming, Southern Ocean phytoplankton growth conditions are anticipated to improve with rising temperatures and a prolonged growing season, as well

as improving light conditions with increasing stratification, shoaling mixed layer depths, and reduced sea ice cover (Bopp et al., 2001; Sarmiento et al., 2004b). Apart from the prominent limitation by light, changes in iron limitation in the Southern Ocean with climate change remain uncertain as the iron cycle is not well resolved mechanistically in Earth System Models (Tagliabue et al., 2016). In addition to bottom-up control, top-down control via zooplankton also affects phytoplankton variability in the Southern Ocean (Le Quéré et al., 2016).

As phytoplankton in the Southern Ocean plays an important role for the local food web and also in setting the biogeochemical characteristics of ocean waters globally, better understanding and constraining its evolution under climate change is relevant for reliable projections of marine biological production well beyond the Southern Ocean (Sarmiento et al., 2004a; Marinov et al., 2006; Nissen et al., 2021). Here, we deliver crucial insights into phytoplankton projections under climate change, highlighting the dynamic interplay between bottom-up and top-down controls due to mixed layer shoaling in the Southern Ocean. By employing an emergent constraint approach, we enhance our confidence in comprehending phytoplankton dynamics and their response to climate change.

## 2 Methods

### 2.1 Research area

This study focuses on the subantarctic and subpolar Antarctic regions, which approximately span the latitude range of $40°\text{S}-60°\text{S}$ (boxes in panel o of Fig. 1 & 2). This region has high levels of macronutrients sufficient to support the growth of phytoplankton and is relatively unaffected by sea ice. In the current climate, the seasonal change in mixed layer depth (MLD) is known to be important for seasonal plankton dynamics, primarily because of its impact on light regulation Arteaga et al. (2020).

### 2.2 Multi-Model Ensemble

Here we use the output of a Multi-Model Ensemble (MME; Table 1) of 14 Earth System Models (ESMs). Of the 14 models that we used, 13 ESMs are part of the Coupled Model Intercomparison Project Phase 6 (CMIP6, downloaded from https://esgf-data. dkrz.de). The FOCI (Flexible Ocean and Climate Infrastructure) model run at GEOMAR (Chien et al., 2022) was added to these 13 models. The selection of these models is based on the present availability of the variables required (phytoplankton biomass concentration, zooplankton biomass concentration, mixed layer depth and surface chlorophyll concentration), the temporal resolution (monthly), and the experimental settings (historical simulation, piControl simulation to account for potential drifts, and the SSP5-8.5 high emission no mitigation scenario). The models within the MME are structurally different, cover a range of different parameterizations of processes, and initial conditions (Séférian et al., 2020). All output fields were regridded on a regular $1° \times 1°$ map using the bilinear interpolation of CDO (Climate Data Operators).

## Relative changes of mixed layer depth

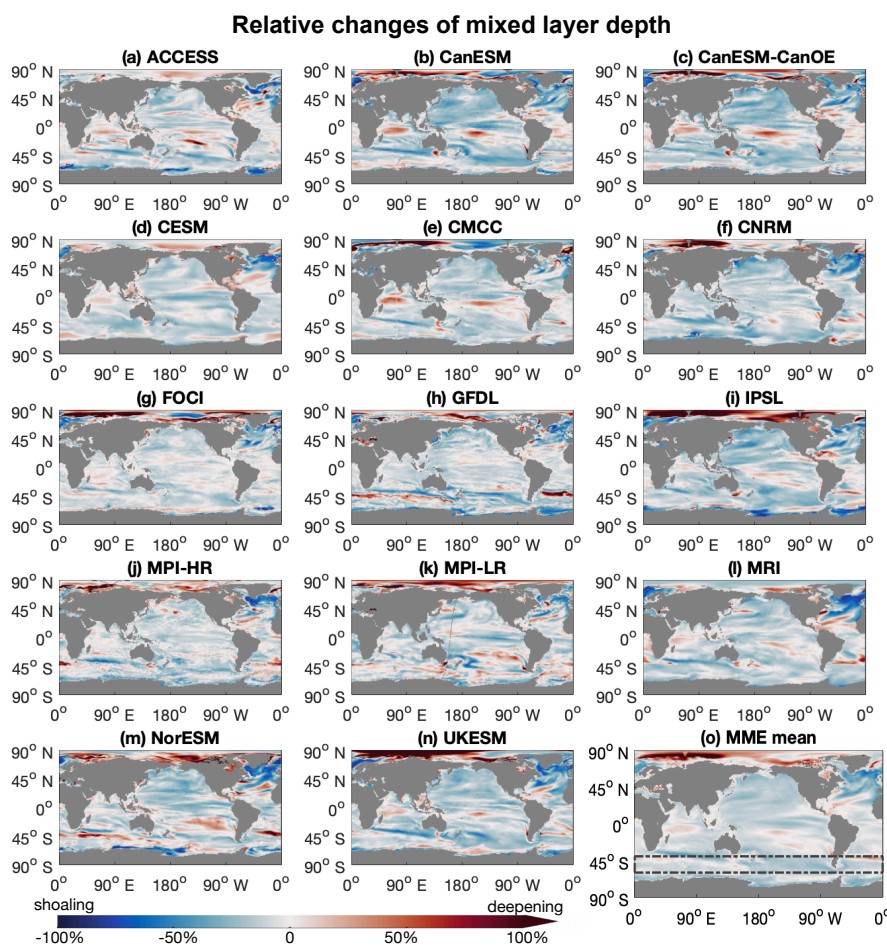

**Figure 1. The mixed layers in the Southern Ocean are projected to shoal under a high emission, no mitigation scenario (SSP5-8.5).** (a-n) Maps for individual climate models in a multi-model ensemble (MME; listed in Table 1), showing annual mean mixed layer depth anomalies of the last decade (2090s) of the 21$^{st}$ century relative to the first decade (2000s), under the SSP5-8.5 scenario. (o) is the mean of the multi-model ensemble. Boxes denoted by dashed lines mark the focus area. Red indicates deepening, and blue indicates shoaling mixed layers towards the end of the 21$^{st}$ century.

**Relative changes of surface phytoplankton concentration**

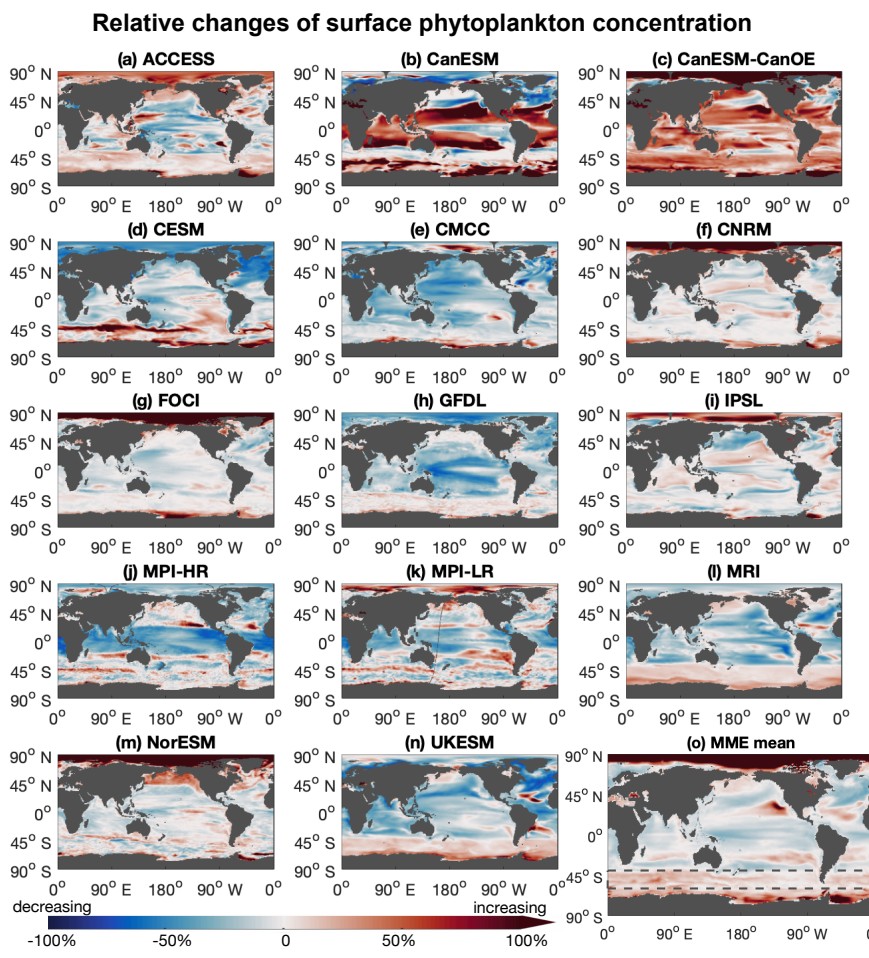

**Figure 2. The Southern Ocean stands out as a region with projected increasing surface phytoplankton concentrations under a high emission, no mitigation scenario (SSP5-8.5).** (a-n) Maps for individual climate models in a multi-model ensemble (MME; listed in Table 1), showing the relative change of surface phytoplankton of the last decade (2090s) of the 21st century relative to the first decade (2000s), under the SSP5-8.5 scenario. (o) is the mean of the multi-model ensemble. Boxes denoted by dashed lines mark the focus area. Red indicates increasing and blue indicates decreasing surface phytoplankton during the 21st century.

**Table 1.** Overview of the Multi-Model Ensemble (MME) used in this study.

| Number | Model Name | Ocean Biogeochemistry | Plankton Groups | Simulations biomass | production | Reference |
|--------|-----------|----------------------|-----------------|---------|------------|-----------|
| (a) | ACCESS-ESM1-5 | WOMBAT* | P1Z1 | (✓) | | Ziehn et al. (2020) |
| (b) | CanESM5 | CMOC | P1Z1 | ✓ | | Swart et al. (2019) |
| (c) | CanESM5-CanOE | CanOE* | P2Z2 | ✓ | ✓ | Christian et al. (2022) |
| (d) | CESM2-WACCM | MARBL* | P3Z1 | ✓ | | Danabasoglu et al. (2020) |
| (e) | CMCC-ESM2 | BFM v5.2* | P2Z2 | ✓ | | Lovato et al. (2022) |
| (f) | CNRM-ESM2-1 | PISCESv2-gas* | P2Z2 | ✓ | ✓ | Séférian et al. (2019) |
| (g) | FOCI1 | MOPS | P1Z1 | ✓ | ✓ | Chien et al. (2022) |
| (h) | GFDL-ESM4 | COBALTv2* | P3Z3 | ✓ | ✓ | Dunne et al. (2020) |
| (i) | IPSL-CM6A-LR | PISCES-v2* | P2Z2 | ✓ | | Boucher et al. (2020) |
| (j) | MPI-ESM1.2-HR | HAMOCC6* | P2Z1 | ✓ | ✓ | Müller et al. (2018) |
| (k) | MPI-ESM1.2-LR | HAMOCC6 * | P2Z1 | ✓ | ✓ | Mauritsen et al. (2019) |
| (l) | MRI-ESM2.0 | NPZD | P1Z1 | ✓ | | Tsujino et al. (2017) |
| (m) | NorESM2-MM | iHAMOCC* | P2Z1 | ✓ | | Tjiputra et al. (2020) |
| (n) | UKESM1-0-LL | MEDUSA * | P2Z2 | ✓ | ✓ | Sellar et al. (2019) |

\* indicates models that explicitly include iron limitation.

PxZy in column "Plankton Groups" indicates x phytoplankton and y zooplankton groups in the respective model.

Column "Simulations" indicates models whose output was used in biomass or production relevant calculations.

(✓) indicates model provides only surface value of phytoplankton biomass as output, therefore it is not used in total plankton biomass analysis in section 3.1.

## 2.3 Emergent constraint

The emergent constraint approach is based on an identifiable relationship in a model ensemble between an observable contemporary sensitivity and an uncertain condition for future climates (Hall et al., 2019; Williamson et al., 2021). The observable contemporary condition can then be used to constrain future model projections, such as previously done for the ocean carbon sink (Kessler and Tjiputra, 2016; Terhaar et al., 2021a, 2022), ocean acidification (Terhaar et al., 2020, 2021b), or marine primary productivity in the tropics (Kwiatkowski et al., 2017). Emergent constraints are often built upon relationships between the same variable at different times, e.g., Kwiatkowski et al. (2017) establish a link between the change in tropical primary production in response to temperature changes on interannual timescales and the change in tropical primary production in response to temperature changes over the 21[st] century. However, emergent constraints can also be built on relationships between different variables if these are mechanistically related, e.g., Terhaar et al. (2021a) used Southern Ocean sea surface salinity to constrain future uptake of anthropogenic carbon in that region because sea surface salinity determines sea surface density and hence the amount of mode and intermediate water formation.

Here, we apply the emergent constraint approach to reduce uncertainties of the projections of the long-term sensitivity of surface phytoplankton concentrations in the Southern Ocean to changes in the mixed layer depth ($S_{clm}$, Eq. 2, period: 2000 - 2099)

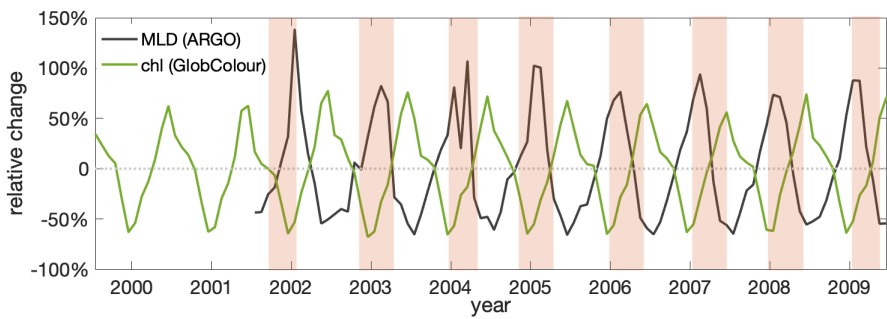

**Figure 3. Observed surface chlorophyll (chl) concentration shows a clear seasonal cycle that, to first order, is anticorrelated with observed mixed layer depths (MLD).** Contemporary seasonal variations relative to the respective annual means of observed MLD (ARGO: black) and surface chlorophyll concentration based on merged satellite data (GlobColour: green) in the Southern Ocean subantarctic and subpolar Antarctic regions ($40^{o}$S$-60^{o}$S). The shaded areas indicate the seasonal MLD shoaling period.

using contemporary seasonal sensitivities of surface chlorophyll concentrations, an observable proxy for surface phytoplankton concentrations with good spatial-temporal coverage from satellite-based sensors, to mixed layer depth changes ($S_{seas}$, Eq. 1, period: 2000 - 2009). The emergent constraint we utilised here, incorporating distinct observable and constraint variables, hinges on the relationship between surface chlorophyll concentration and surface phytoplankton biomass concentration across individual models (Fig. A2). Benefiting from its comprehensive spatial-temporal coverage, chlorophyll provides an ideal balance by offering a strong linear correlation, easy observational access, and comparatively low observational uncertainties. For each of the individual models and for the observations, the respective contemporary seasonal sensitivity of surface chlorophyll concentrations to the MLD ($S_{seas}$) is computed as follows:

$$S_{seas} = \frac{(chl_{mmax} - chl_{mmin})/\overline{chl}}{(MLD_{mmax} - MLD_{mmin})/\overline{MLD}}, \tag{1}$$

with overlines indicating the annual mean and $mmax$ and $mmin$ indicating the months with the minimum and maximum MLD that mark the beginning and the end of the shoaling period (shaded periods in Fig. 3). We explicitly selected the shoaling period, as the decrease due to MLD shoaling and improving light conditions are analogues to changes expected under global warming.

Based on the strong correlation between projected surface chlorophyll and surface phytoplankton (Fig. A2), the long-term sensitivities of surface phytoplankton concentrations to mixed layer changes ($S_{clm}$, Eq. 2) of individual models are calculated based on the relative changes in surface phytoplankton concentration ($R_{sphy}$) and MLD ($R_{MLD}$) between the first decade (2000s) and the last decade (2090s) of the 21$^{st}$ century. The long-term sensitivity calculation here is following the definition of the seasonal sensitivity in Eq. 1.

$$S_{clm} = \frac{(\overline{sphy_{2090s}} - \overline{sphy_{2000s}})/\overline{sphy_{2000s}}}{(\overline{MLD_{2090s}} - \overline{MLD_{2000s}})/\overline{MLD_{2000s}}} = \frac{R_{sphy}}{R_{MLD}} \tag{2}$$

Relative MLD anomalies are used instead of absolute anomalies, as the relative anomaly relates to the dilution of phytoplankton across the mixed layer, which is one of the main mechanisms controlling surface phytoplankton concentration. Despite the large differences revealed in the absolute MLD and chlorophyll values of individual models (Fig. A3), the models resolve the same seasonal anticorrelation between relative variations of MLD and chlorophyll as seen in observations (see results section 3.3.1).

The probability density functions (PDFs) of the long-term sensitivity ($S_{clm}$) were derived from a Gaussian distribution and the MME mean and standard deviation, assuming that all models have equal probability. We calculated the constrained PDFs following previous literature (Eq. 3; Eq. 20 in Williamson et al., 2021).

$$p(y) = \int_{-\infty}^{\infty} p(y \mid x)p(x)dx = \mathcal{N}\left(y \mid a + bX_{obs}, \sqrt{\sigma_f{}^2 + b^2\sigma_{obs}{}^2}\right) \tag{3}$$

The mean of the distribution is calculated by applying the observational constraint ($X_{obs}$) to the emergent linear inter-model relationship ($y = a + b * x$) of $S_{seas}$ and $S_{clm}$. The standard deviation of the PDF distribution is estimated by accounting for both observational uncertainty ($\sigma_{obs}$) and the prediction interval of the emergent relationship ($\sigma_f$).

The hereby constrained $S_{clm}$ (here defined as $S_{clm*}$) is then used to estimate the constrained surface phytoplankton concentration change by the end of the 21st century ($R_{sphy*}$) by reordering Eq. 2 to $R_{clm*} = R_{MLD} * S_{clm*}$. The uncertainty in the post-constraint projection for the relative surface phytoplankton concentration variation is calculated by accounting for uncertainties from both the constrained long-term sensitivity ($\sigma_{S_{clm*}}$) and the MLD projection ($\sigma_{R_{MLD}}$) through error propagation:

$$\mathcal{N}\left(R_{sphy*}, \sigma_{sphy*}\right) = \mathcal{N}\left(R_{MLD} \cdot S_{clm*}, |R_{MLD} \cdot S_{clm*}| \sqrt{\left(\frac{\sigma_{R_{MLD}}}{R_{MLD}}\right)^2 + \left(\frac{\sigma_{S_{clm*}}}{S_{clm*}}\right)^2}\right) \tag{4}$$

## 2.4 Observational constraints

The observation-based estimate of $S_{seas}$ is based on monthly MLD data from the ARGO mixed layer database (http://mixedlayer.ucsd.edu/) and satellite-derived monthly surface chlorophyll data from the merged satellite product GlobColour (https://www.globcolour.info/) during the period 2000 - 2009. The observational MLD is calculated based on the temperature threshold estimate (threshold of 0.2°C; Holte and Talley, 2009), consistent with the MLD diagnostic available from the MME. Surface chlorophyll is used as a reliable proxy for surface phytoplankton under the current climate, as it is a comparatively well-observed variable. Under the common assumption that all properties (except light) are considered homogeneous within the mixed layer (Westberry et al., 2008), we assume that phytoplankton concentrations across the mixed layer are well reflected by surface phytoplankton concentration. The seasonal variation in the observability of surface chlorophyll from satellite data

is mainly affected by the winter light conditions and varying ice cover to the south of the research region. The impact of cloud
cover in the research region is minor at a monthly temporal resolution (Fig. A4).

## 2.5  Trophic transfer efficiency

Given the simplicity of models and the limitations of their outputs, the trophic transfer efficiency (TTE, Eq. 5) is simply defined
as the ratio of total (depth-integrated) biomass of zooplankton to that of phytoplankton following Barnes et al. (2010),

$$TTE = \frac{\sum_{z=0}^{zmax} zoo * dh}{\sum_{z=0}^{zmax} phy * dh} \qquad (5)$$

where phy and zoo represent the phytoplankton and zooplankton biomass concentrations (unit: $mg\ C\ m^{-3}$), respectively; dh
indicates the thickness of each grid box, and zmax the depth of the water column.

The dimensionless TTE reflects the biomass or energy transfer between trophic levels and is typically estimated based on
the biomasses or productivities of trophic levels. We used biomass instead of productivity to quantify the TTE, as biomass can
be directly compared to observations and is available from more models than productivity data (Table 1).

## 3  Results

### 3.1  Stable phytoplankton biomass in a changing climate: shifting balance of bottom-up and top-down control in the Southern Ocean

The multi-model ensemble (MME) projects the total phytoplankton biomass to remain relatively stable (-2 ± 8%) through-
out the 21$^{st}$ century, following the SSP5-8.5 scenario (Fig. 4a). Future changes in total phytoplankton biomass are largely
determined by the competing effects of increasing primary production (bottom-up, Fig. 4b) and grazing by zooplankton (top-
down, Fig. 4c). Understanding the complex interplay between bottom-up and top-down controls is crucial for projecting how
phytoplankton will respond to future climate change.

Projected increases in total primary production (10 ± 11%) show that phytoplankton growth conditions are improving with
climate change, indicating weakening bottom-up control (Fig. 4b). On these centennial timescales, climate change-induced
changes in primary production in the subantarctic and subpolar Antarctic regions are tightly coupled to MLD-driven bottom-
up changes (e.g., light, nutrients and temperature) that also drive changes on observed contemporary seasonal timescales
(Arteaga et al., 2020; Leung et al., 2015). As the MLD is projected to shoal by 12 ± 5% over the course of the 21$^{st}$ century
(Fig. 5a), phytoplankton in the water column will largely stay closer to the surface, not being passively mixed down, thereby
experiencing better light and temperature conditions. As a result, phytoplankton are projected to have a higher growth rate in
the nutrient-replete Southern Ocean (dashed line in Fig. 4b).

Despite increasing total primary production, the total phytoplankton biomass remains relatively stable throughout the 21$^{st}$
century mainly because of enhanced zooplankton grazing (14 ± 15%), thus intensified top-down control (Fig. 4c). With top-
down control exhibiting a similar increasing trend as bottom-up condition under climate change, phytoplankton biomass stays

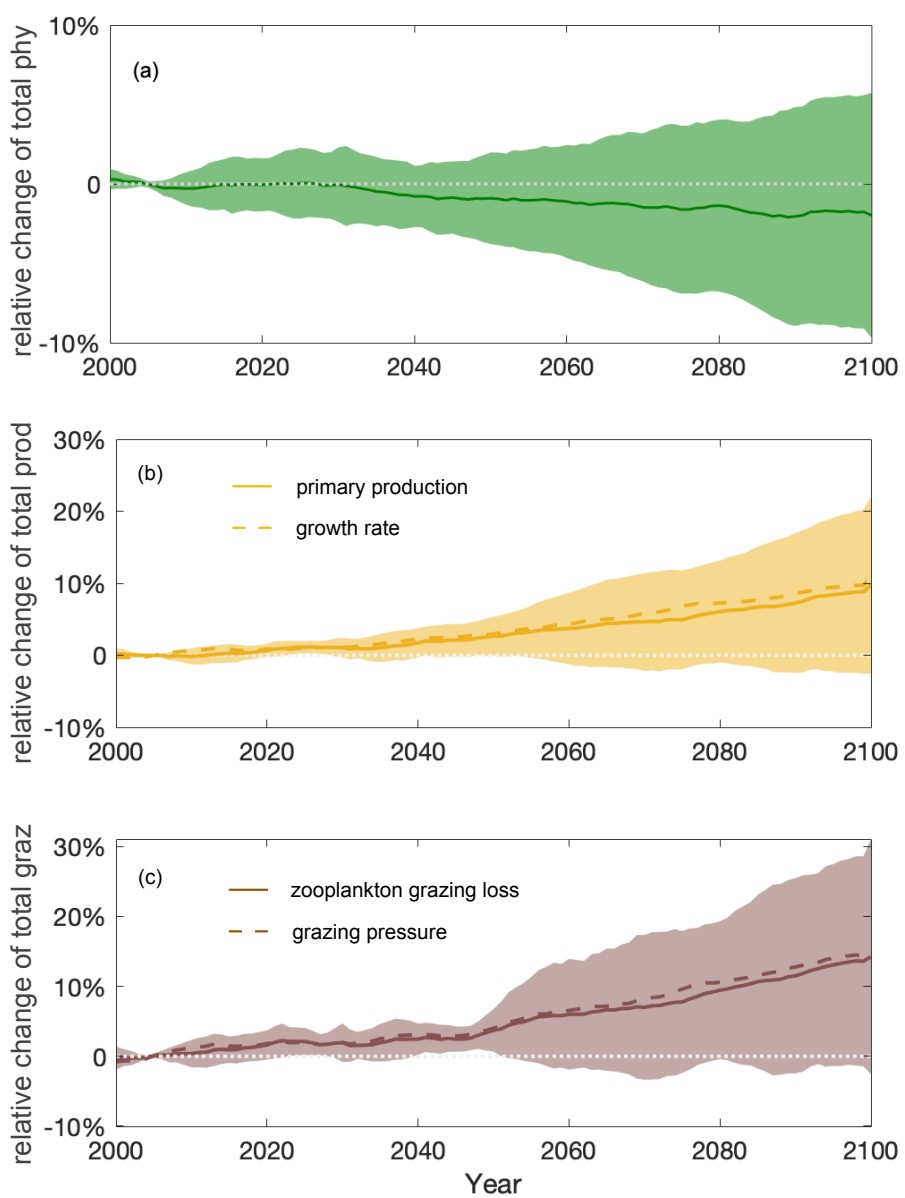

**Figure 4. Stable total phytoplankton projections over the course of the 21st century due to a shifting balance of bottom-up and top-down control.** Multi-model ensemble (MME) projections of the relative changes of total (a) phytoplankton (phy, dark green), (b) phytoplankton production (prod, yellow) and (c) grazing loss from zooplankton (graz, brown) from 2000 to 2100 under the SSP5-8.5 scenario, with shading indicating one standard deviation across the ESMs, relative to the respective mean values of the first decade of the 21st century (2000 - 2009) in the Southern Ocean. Solid lines indicate the multi-model mean. Dashed lines in (b) and (c) indicate the MME averaged phytoplankton growth rate (estimated for individual models as integrated production divided by integrated biomass) and grazing pressure (estimated as integrated grazing loss divided by integrated biomass), respectively. The time series are filtered using a 10-year moving average.

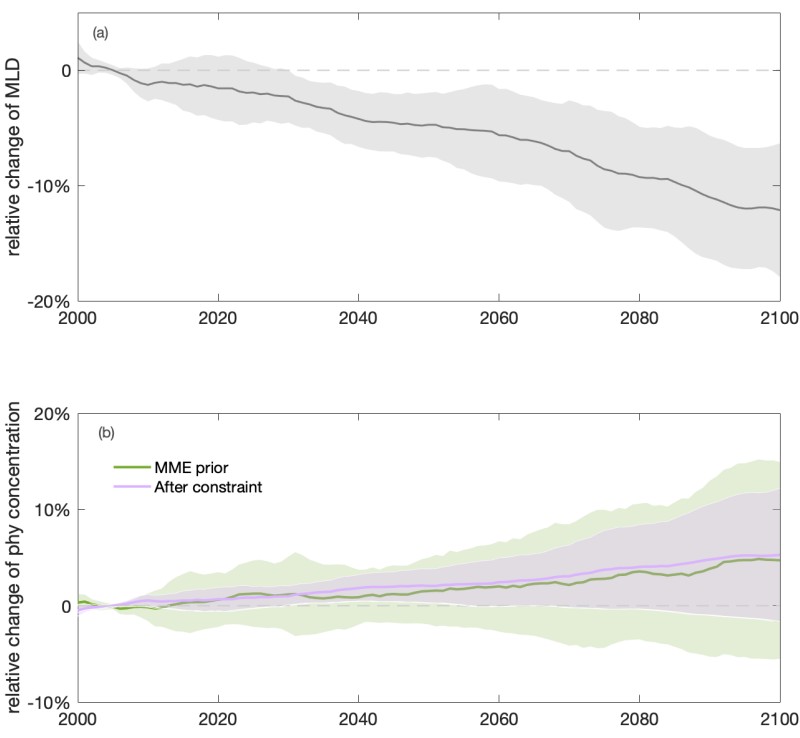

**Figure 5. Shoaling of annual mean mixed layer depth (MLD) and increasing annual mean surface phytoplankton concentration towards the end of the century in the Southern Ocean.** Multi-model ensemble (MME) projections of changes of annual mean (a) MLD (grey); and (b) surface phytoplankton concentration (sphy) before (light green) and after (purple) applying the emergent constraint, from 2000 to 2100, relative to the mean values of the first decade of the 21[st] century (2000 - 2009), with shading indicating one standard deviation, under the SSP5-8.5 scenario. The time series are filtered using a 10-year moving average.

in a dynamic balance despite the flux changes, and thereby appears not to respond to the changing climate. An enhanced role
of grazing under climate change is consistent with findings in Laufkötter et al. (2015).

### 3.2   Increasing surface phytoplankton concentration: intensified top-down control shifts trophic structure

The increasing top-down control and grazing pressure on phytoplankton may be a consequence of rising phytoplankton concentrations due to mixed layer shoaling (Fig. 5 a & b) and higher temperatures (Fig. A5) under climate change. Apart from total phytoplankton biomass, which reflects the total amount of food available to zooplankton, the concentration of phytoplankton
is also a crucial factor in zooplankton grazing, as the feeding efficiency of zooplankton depends directly on the concentration of phytoplankton. Increasing phytoplankton concentration, due to shoaling mixed layer depths, enhances encounter efficiency

between zooplankton (the predator) and phytoplankton (the prey), enabling more efficient grazing (dashed line in Fig. 4c) and thus a stronger top-down control on phytoplankton. This mechanism is apparent in the correlation between mixed layer depth and phytoplankton grazing loss due to zooplankton on the long-term time scale (Fig. A6) and has previously been found on a seasonal scale in present-day productive systems (Xue et al., 2022a). Additionally, higher temperatures are associated with increased zooplankton grazing rates and, thereby, a stronger top-down control on phytoplankton (Caron and Hutchins, 2013), though only a few CMIP6 models include temperature-dependent grazing (Rohr et al., 2023).

Increased top-down grazing pressure results in higher trophic transfer efficiency by the end of the century (4 $\pm$ 7%; Eq. 5, Fig. 6b). The rising trophic transfer efficiency indicates a potential change in trophic structure, with relatively more zooplankton and less phytoplankton in the future climate. In contrast to the projection of slightly decreasing total phytoplankton biomass (-2 $\pm$ 8%), the total biomass zooplankton, a phytoplankton predator, is projected to increase by 2 ($\pm$11)% under climate change (Fig. 6a). Zooplankton appears to be favoured by changing environmental conditions and food resources and is grazing down phytoplankton. The phenomenon of a positive change on a higher trophic level and a negative change on the lower trophic level indicates increasing top-down control of the lower trophic level by the higher one (Fig. 1 in Chust et al., 2014).

## 3.3 Constraining the increasing surface phytoplankton concentration: the underlying mechanism of intensified top-down control

### 3.3.1 Opposing trends of surface phytoplankton concentrations and mixed layer depths on timescales of seasons and of climate change

As phytoplankton concentration is an indicator of grazing pressure, the large inter-model differences in projected changes from the 2000s to the 2090s (5 $\pm$ 10%; Fig. 5b) cause a large part of the inter-model differences of projected changes in phytoplankton top-down control. Individual model projected surface phytoplankton concentration changes range from -10% to +32% and in particular include zero and negative changes in the Southern Ocean by the end of the century. The mechanisms previously suggested to explain the long-term changes in phytoplankton, such as light and temperature changes, are also responsible for their seasonal variability in the Southern Ocean (Leung et al., 2015; Henley et al., 2020).

Under the current climate, seasonalities of mixed layer depths and surface chlorophyll concentration are anti-correlated in both observations and model simulations. The observed surface chlorophyll in the Southern Ocean exhibits a clear seasonal cycle, peaking in late austral spring and early austral summer (Fig. 7). This seasonal cycle of chlorophyll is opposite that of mixed layer depth (MLD), with shallow MLD in austral summer coinciding with relatively high surface chlorophyll, and deep MLD in austral winter coinciding with relatively low surface chlorophyll. Such an opposite relationship between surface chlorophyll and MLD on a seasonal scale has been previously shown in observations (Uchida et al., 2019; Arteaga et al., 2020) and model simulations (Song et al., 2018; Arteaga et al., 2020; Le Quéré et al., 2016). This relationship is thought to be mainly caused by the seasonal dilution of phytoplankton and by growth limitation along with zooplankton grazing. MLD reveals higher uncertainty compared to the extensively covered observed chlorophyll (satellite-based estimates), likely due to the scarcity of in situ-based data used to estimate the MLD. The individual models largely agree with the observed seasonal

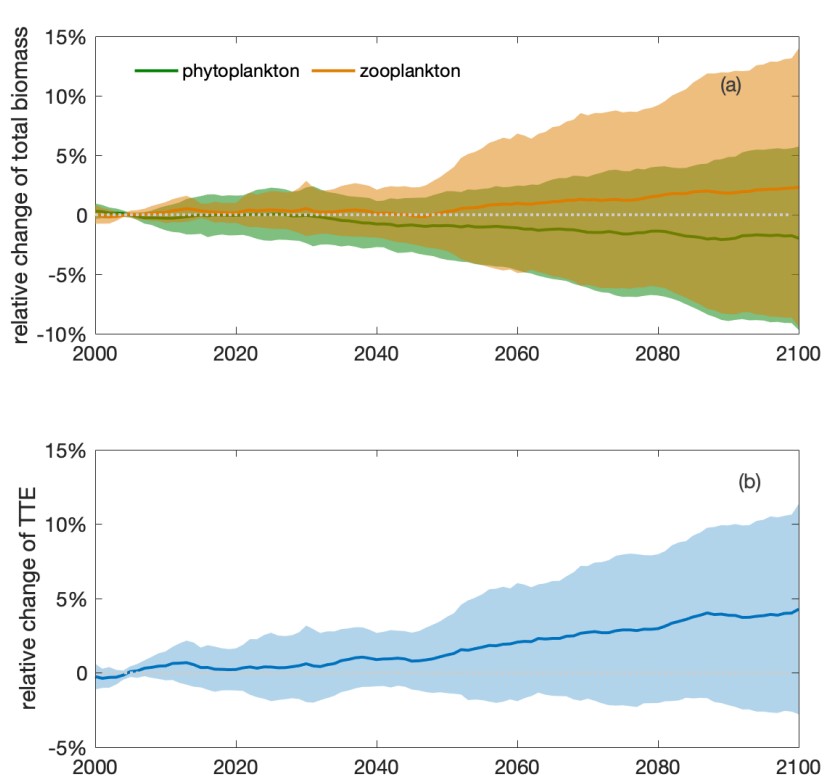

**Figure 6. An increase in total zooplankton biomass and trophic transfer efficiency (TTE) during the 21st century indicates intensified top-down control and a shift in trophic structure.** Multi-model ensemble (MME) projections of the relative changes of (a) total (depth-integrated) phytoplankton (green) and zooplankton (orange) biomasses; and (b) trophic transfer efficiency (blue; defined as the ratio of the two, see Eq. S3) from 2000 to 2100 under the SSP5-8.5 scenario, with shading indicating one standard deviation, relative to the respective mean values of the first decade of the 21st century (2000 - 2009) in the Southern Ocean. The time series are filtered using a 10-year moving average.

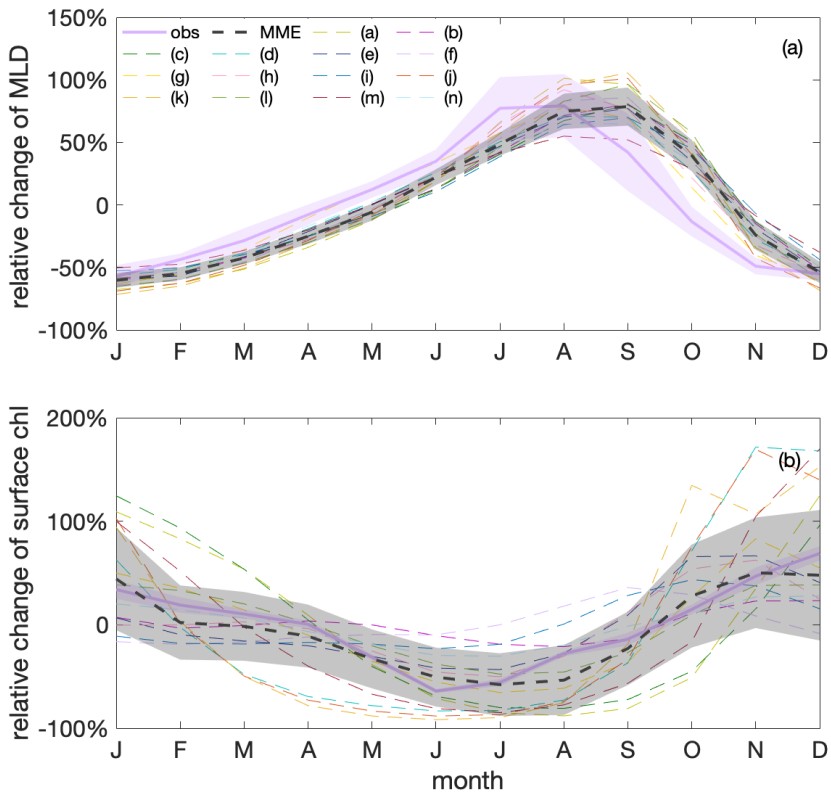

**Figure 7. Surface chlorophyll (chl) concentration shows a clear seasonal cycle that is anticorrelated with mixed layer depths (MLD) in both observations and model simulations.** Seasonal cycles of (a) MLD and (b) chl variations relative to their respective annual mean values, based on observations (GlobColour, purple) and a multi-model ensemble (MME, grey); the black dashed line marks the MME median value, and the shadings indicate the standard deviations of observations and MME, respectively. Differently coloured thin dashed lines show individual models listed in Tab. 1 that contribute to the MME median value.

cycles, though with some spread (Fig. 7). Such spread is not unexpected given the structural differences of the biogeochemical models that include varying considerations and representations of the chlorophyll to carbon (chl:C) ratio, grazing, and nutrient limitations (such as iron). On average, the modelled MLD maximum shows a lag of around a month compared to observational estimates, which is reflected in a similar lag in the chlorophyll seasonality. In both, observations and models, there is a lag between the timing of the peaks of minimum chlorophyll and maximum MLD in winter. This lag is likely due to poor light

conditions in deep MLDs triggering an increase in the chl:C ratio, where increasing chlorophyll pigments mask a continued decrease in phytoplankton biomass (organic carbon) concentration in deepening MLDs (Geider, 1987; Arteaga et al., 2016).

### 3.3.2 Using contemporary seasonality to constrain projections of surface phytoplankton concentration

Despite a good understanding of the top-down and bottom-up mechanisms in ESMs in the Southern Ocean, as explained above, the inter-model uncertainty of these mechanisms remains large. Therefore, we employ an emergent relationship between surface phytoplankton biomass and mixed layer depth to reduce these uncertainties. A linear relationship ($r^2$=0.65, Fig. 8a) exists across the ESM ensemble between this sensitivity of chlorophyll to changes in MLD on a seasonal scale in the current climate (seasonal sensitivity, $S_{seas}$, Eq. 1) and the sensitivity of surface phytoplankton to changes in MLD with climate change (climate change sensitivity, $S_{clm}$, Eq. 2), as expected based on previous studies (Leung et al., 2015; Henley et al., 2020). The relationship suggests that models within which surface chlorophyll is more sensitive to MLD changes on a seasonal scale also tend to show a larger sensitivity on a longer-term scale. Moreover, the slope of the emergent relationship is nearly one. A one-to-one relationship indicates the same sensitivities of the leading mechanism on different timescales (Williamson et al., 2021). Though the sensitivities of chlorophyll to changes in MLD on a seasonal scale from individual models show some spread, it is important to note that the models deviating from the observed sensitivity are still considered capable of representing the relationship between chlorophyll sensitivity to MLD changes across seasonal and long-term scales. The correlation between mixed layer depth and surface phytoplankton on a long-term climate scale is consistent with reduced dilution and a weakening of bottom-up controls, analogous to the most important mechanisms on seasonal scales: when the MLD is shoaling and phytoplankton are being contained in a shallower surface mixed layer with higher average light intensity and temperature, as a result, phytoplankton grow better and concentrations are expected to increase (Xue et al., 2022b). With this, we can use the relationship to constrain the projection of surface phytoplankton concentration under climate change.

Applying the emergent constraint derived from the observations under current seasonality confirms the increasing trend of surface phytoplankton concentration by the end of the 21[st] century (Fig. 5b), thereby indicating intensified top-down control, with higher confidence. Utilising the observational constraint on the seasonal sensitivity $S_{seas}$, the emergent constraint (section 2.3) adjusts the estimate of the long-term sensitivity ($S_{clm}$) from -0.38 ($\pm$ 0.69) to -0.44 ($\pm$ 0.46) (Fig. 8b). Thus, surface phytoplankton concentration becomes slightly more sensitive to MLD shoaling than previously suggested by the unconstrained MME results. In combination with the MME projected MLD change, our emergent constraint implies a 5 ($\pm$ 7)% (unconstrained: 5 $\pm$ 10%) increase in surface phytoplankton concentrations in the Southern Ocean by the end of the 21[st] century (Eq. 4). The emergent constraint reduces the uncertainty in the surface phytoplankton projection by 34% (Fig. 5b). Based on the constraint, the possibility of a surface phytoplankton concentration decrease becomes less likely (16%) compared to what was expected based on the MME alone (29%). Our results therefore suggest, with enhanced confidence, an increase in surface phytoplankton concentration in the subantarctic and subpolar Antarctic regions of the Southern Ocean by the end of the century (Fig. 5). This projection of increasing surface phytoplankton concentration, alongside rising temperatures (Fig. A5), suggests an increased likelihood of enhanced grazing pressure and thus a more intensified top-down control under climate change.

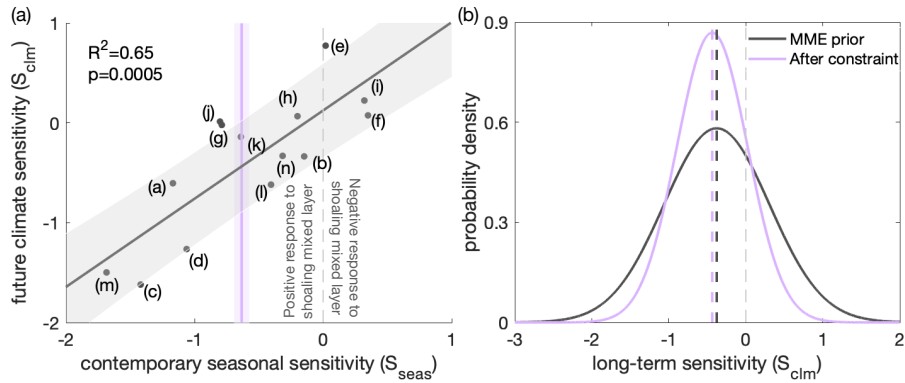

**Figure 8. Application of an emergent constraint to the long-term sensitivity ($S_{clm}$) of surface phytoplankton concentration to mixed layer depth (MLD), based on the contemporary seasonal sensitivity ($S_{seas}$) of surface chlorophyll concentration to MLD.** (a) Long-term sensitivity ($S_{clm}$, eq. 2) under climate change against contemporary seasonal sensitivity ($S_{seas}$, eq. 1) of the relative change of surface chlorophyll (for $S_{seas}$) or phytoplankton (for $S_{clm}$) concentration to MLD for the multi-model ensemble (MME, black dots, corresponding to the models listed in Table. 1). The fit of a linear regression (black line) and the associated 68% prediction intervals are shown as a black line and grey shading, respectively. The purple vertical line indicates an observational constraint, with purple shading indicating the associated uncertainty, calculated as the standard error of the climatological $S_{seas}$ of the observed time series (Fig. 3). The light grey dashed line separates the positive (to the left) and negative (to the right) responses across models of surface chlorophyll to MLD shoaling under present-day seasonality. (b) Probability density functions (PDFs) of $S_{clm}$. The black line shows the "MME prior" PDF, assuming all model projections are equally likely and come from a Gaussian distribution. The purple line shows the observationally constrained PDF (it has a similar mean but is narrower). Dashed lines indicate the mean values of the future climate sensitivity before (black) and after applying the constraint (purple).

## 4 Discussion

### 4.1 Phytoplankton under climate change

In this study, we have delved into the complex interplay of bottom-up and top-down mechanisms affecting phytoplankton dynamics in the Southern Ocean under climate change. Building on the understanding of these mechanisms across current seasonal and future climate change scales, we are employing emergent constraints to refine surface phytoplankton projection and emphasising the weakening bottom-up control and the growing importance of top-down control. This approach not only advances our understanding of the Southern Ocean's ecosystem response to climate change but also illustrates the potential for
similar applications in other productive oceanic regions with notable model uncertainties, such as coastal upwelling regions (Chang et al., 2023).

While emergent constraints enable MME to project future phytoplankton changes with higher confidence, the often too simplistic representation of the iron cycle in ESMs (Tagliabue et al., 2017) may hide additional uncertainties and the oversimplified food web formulations potentially limit the reproduce of phytoplankton composition changes, underscoring the need

for model improvements to capture these critical changes and understand future ecosystem dynamics comprehensively (Petrou et al., 2016; Krumhardt et al., 2022). In addition to impacts from light and temperature change, the micronutrient iron is often deemed limiting in the Southern Ocean, which affects phytoplankton growth and limits the magnitude of phytoplankton blooms (Martin et al., 1990; Moore et al., 2013; Arteaga et al., 2020). Over recent decades, an increase in iron stress has been observed in the Southern Ocean due to changing mixed layer dynamics (Ryan-Keogh et al., 2023). However, with the progression of climate change, an increase in the atmospheric iron supply, a crucial source for the Southern Ocean, is projected to rise through various sources (Pörtner et al., 2019; Hamilton et al., 2020). For instance, the deposition of soluble iron is expected to increase due to more frequent and intense wildfires (Bowman et al., 2020), along with increasing wind and desertification (Woodward et al., 2005). Despite its importance to phytoplankton growth in the Southern Ocean, the processes of producing and cycling iron are still not yet fully understood, and it was not until the early 2000s that global ocean models began incorporating iron (Tagliabue et al., 2017; Moore et al., 2001). Even in the MME used here from the most recent generation of Earth system models, 3 out of 14 ESMs do not represent the iron cycle at all (Table 1). Of the 11 ESMs that include the iron cycle, only 9 models provide output on iron, simulating varying changes in iron availability in the Southern Ocean. However, across the multi-model mean, there is no discernible trend in surface iron concentration (Fig. A7). As a result, models with or without iron limitation do not reveal clear differences in the projection of phytoplankton responses to climate change (Fig. A8), though models with iron representation show a much larger spread than those without. Given the very different representation of the iron cycle in current ESMs and the complex interplay between iron and its biological responses, along with the multitude of external processes affecting its availability, projecting changes in iron availability likely adds a large uncertainty to phytoplankton projections under climate change (Petrou et al., 2016).

## 4.2 Enhancing understanding of top-down processes under climate change

While bottom-up mechanisms are well recognised and studied (Deppeler and Davidson, 2017; Frenger et al., 2018; Arteaga et al., 2020; Leung et al., 2015), the impact of top-down effects on phytoplankton has been less emphasised, largely due to insufficient observational data. Despite significant advancements in ocean ecosystem monitoring over recent decades, such as the COPEPOD dataset (Coastal and Oceanic Plankton Ecology, Production, and Observation Database, https://www.st.nmfs. noaa.gov/copepod/; Moriarty and O'brien, 2013), supporting advanced relevant studies, a notable gap persists in observational data that specifically target higher trophic levels and data coverage in the Southern Ocean. This gap underscores the need for a more comprehensive observational system, especially concerning quantities such as zooplankton biomass, community composition, and traits, to enhance the robustness of projections of future plankton dynamics and trophic transfer (Prowe et al., 2022). It is important to understand zooplankton physiology to identify and further constrain the relevant key parameters, such as prey preferences, maximum ingestion rates, and zooplankton temperature sensitivity (Prowe et al., 2019; Petrik et al., 2022).

Climate change projections of top-down variables are highly uncertain (Laufkötter et al., 2015; Lotze et al., 2019; Kwiatkowski et al., 2020; Rohr et al., 2023), mainly because zooplankton and food web formulations in large-scale models are oversimplified, posing a substantial caveat. The formulation of zooplankton in numerical models is crucial in regulating the mortality of modelled phytoplankton, which in turn has a significant impact on planktonic dynamics (Le Quéré et al., 2016; Prowe et al.,

2019). Top-down grazing by zooplankton is influenced by factors including traits of both prey and predator, prey concentration, and also by the type of predator-prey relationship used in a model, such as the different Holling types (Kiørboe, 2009; Xue et al., 2022a; Anderson et al., 2010). In addition, the lack of observations has hindered the calibration of most global biogeochemical models to accurately represent top-down controls (Stock et al., 2014b). To improve the top-down projection, ocean biogeochemical models should more regularly save zooplankton-related variables as standard model output to allow for comprehensive analyses (Laufkötter et al., 2015; Le Grix et al., 2022). Such zooplankton-related variables should include not only bulk variables such as biomass but also fluxes in order to better track the energy flux through the ecosystem. Overall, model representations of phytoplankton will benefit from a more extensive assessment of zooplankton through a more accurate simulation of the top-down process.

### 4.3 Differences in trends of biomass and production

The divergent trends in total phytoplankton biomass and production under climate change demonstrated in our study are attributed to the increasing importance of top-down control. Here, we show that total phytoplankton biomass remains relatively stable in the Southern Ocean, while total phytoplankton production continues to increase. Previous investigations of the phytoplankton response to climate change focusing on biomass (Marinov et al., 2010; Kwiatkowski et al., 2019; Lotze et al., 2019) versus production (Behrenfeld et al., 2006; Laufkötter et al., 2015; Deppeler and Davidson, 2017; Kwiatkowski et al., 2020) tended to conclude that changes would be of the same sign for the two variables, with a negative response in low-latitude regions due to increasing nutrient limitations and a positive response in high-latitude regions as a result of improving light conditions. Further investigations into the trophodynamics using biomass (Chust et al., 2014; Kwiatkowski et al., 2019) and production (Stock et al., 2014a) have also demonstrated consistent findings, with low-latitude regions exhibiting negative trophic amplification (with decreases in higher trophic levels exceeding those in the lower trophic levels) and high-latitude regions demonstrating positive trophic amplification (with increases in higher trophic levels exceeding those in the lower trophic levels) under climate change. The change in biomass is determined by the competing effects of bottom-up (production) and top-down (loss) processes. Projections using biomass and production tend to yield the same trends when top-down processes are negligible or play a secondary role in comparison with bottom-up processes. However, when top-down processes are of similar magnitude (e.g., Fig. 4) or even more important than bottom-up processes, projections using biomass and production may yield different results as biomass and production represent two distinct ways of viewing and understanding the ecosystem. While biomass focuses on the standing stock that can be measured, production quantifies the energy flow among different components of the ecosystem. Our finding of different trends in phytoplankton biomass and production under global warming underscores the increasing importance of top-down processes in the future Southern Ocean ecosystem.

### 5 Conclusions

In this study, we present the future phytoplankton response to climate change under a high emission, no mitigation (SSP5-8.5) scenario in the subantarctic and subpolar Antarctic regions ($40°$S$−60°$S) of the Southern Ocean using a multi-model ensemble.

We find that the total phytoplankton biomass stays relatively stable over the course of the 21$^{st}$ century as a result of a dynamic balance between bottom-up and top-down control. Future phytoplankton growth conditions are expected to improve as a result of the shoaling mixed layer and increasing temperature, thereby weakening bottom-up control, consistent with previous studies. Meanwhile, the shoaling mixed layer will further concentrate the phytoplankton biomass and make phytoplankton more accessible to zooplankton, thereby intensifying grazing pressure and hence top-down control. The continuously growing negative effect of increasing top-down control compensates for the positive effects of improving bottom-up conditions, resulting in a well-balanced flux budget that keeps the total phytoplankton biomass stable under climate change.

We further employ the approach of an emergent constraint to increase our confidence in the increasing trend of surface phytoplankton concentration, which could be the underlying mechanism that contributes to the intensifying top-down processes under climate change. For that, we make use of the observed relationship between seasonal variations in mixed layer depth and surface chlorophyll concentration. The relationship originates from the combined effects of bottom-up and top-down processes as well as physical dilution caused by mixing across the varying mixed layers. The application of the observational constraint to a climate projection under SSP5-8.5 scenario confirms the increasing trend of surface phytoplankton concentration and reduces the uncertainty of its future projections by more than a third. This further promotes confidence in intensified top-down control in the changing climate.

This study highlights the need to understand the balance of bottom-up versus top-down control of phytoplankton. In particular, mechanisms promoting a dominance of top-down control, which are of growing importance for climate change, have been understudied. Top-down control is typically treated stepmotherly as a mere "closure term" in biogeochemical models for climate research, such as CMIP models, which prioritize numerical stability while often neglecting zooplankton physiology and adaptive capacities. We argue that zooplankton physiology and adaptive capacities are crucial for accurately simulating phytoplankton dynamics. To improve our understanding of the response of phytoplankton to climate change, there is a pressing need for dedicated observational and modelling efforts. Such efforts include, but are not limited to, long-term and continuous observational time series of phytoplankton and zooplankton biomass and traits to enhance our understanding of the response of feeding processes to changing environmental conditions. Such observations, in turn, will facilitate improved top-down formulations in models. Further, our results suggest that previous studies (e.g., Chust et al., 2014; Kwiatkowski et al., 2017; Stock et al., 2014a) that result in the same conclusion for phytoplankton biomass and production imply a negligible or secondary role of top-down processes. With the increasing importance of top-down processes under climate change, phytoplankton biomass and production could display divergent trends, as shown in our results. Our results demonstrate that total phytoplankton biomass in the Southern Ocean can remain stable despite increasing phytoplankton production as a result of a shifting balance from bottom-up towards enhanced top-down control.

*Author contributions.* TX and IF designed the study. TX conducted the analysis with support from IF and JT. TX wrote the first draft of the manuscript and produced all figures and tables. All authors discussed the results and contributed to the final writing of the manuscript.

*Competing interests.* The authors declare that they have no conflict of interest.

*Acknowledgements.* This work is financially supported by the China Scholarship Council (TX, grant no.201808460055). JT acknowledges
funding from the Woods Hole Oceanographic Institution Postdoctoral Scholar Program and founding from the Swiss National Science
Foundation Ambizione project ArcticECO - PZ00P2_209044. TLF acknowledges funding from the European Union's Horizon 2020 research
and innovation programme under grant agreement no. 862923 (project AtlantECO). IF acknowledges funding from BMBF project Humboldt
Tipping II (01LC2323B). We would like to thank Wolfgang Koeve and Markus Schartau for discussions and constructive feedback.

## Appendix A: Additional figures

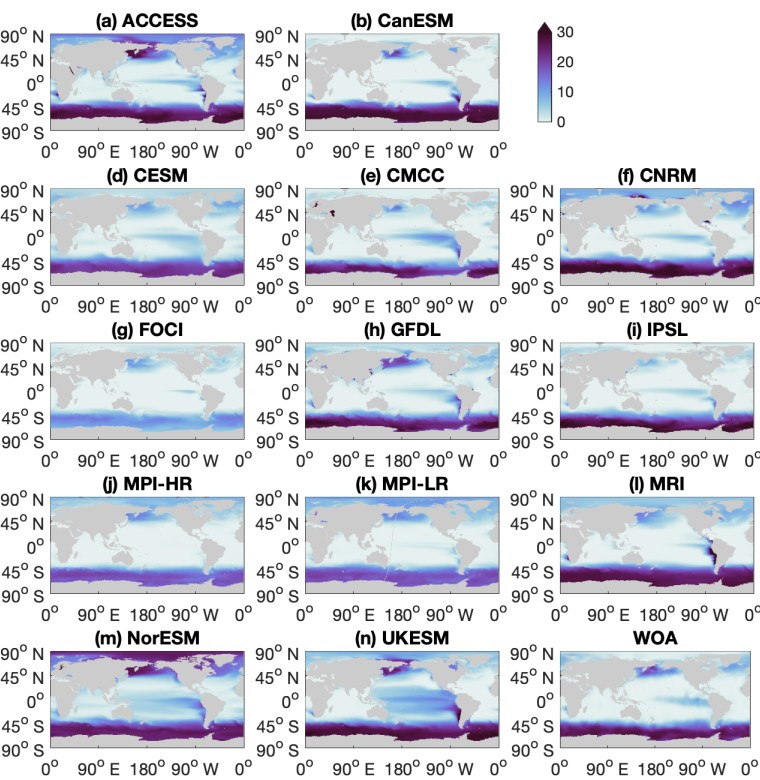

**Figure A1.** Maps for individual climate models and observational data from the World Ocean Atlas, showing annual mean surface nitrate concentration (unit: $mmol\,m^{-3}$) in contemporary climate (2000-2009).

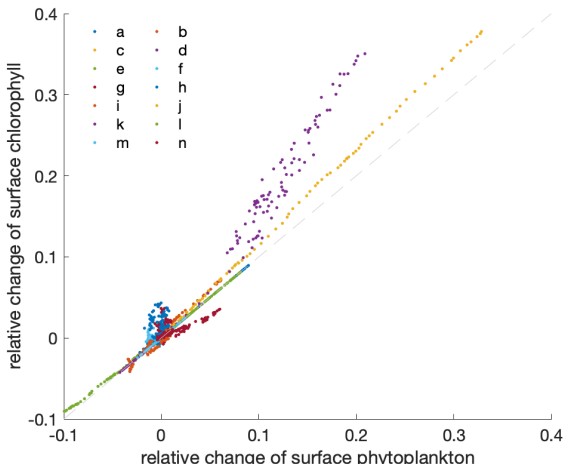

**Figure A2. Relation between the relative changes of annual mean surface phytoplankton concentration and surface chlorophyll concentration.** The light grey dashed line indicates the 1:1 line. Differently coloured dots indicate individual models as listed in Table 1 in the main manuscript, with individual dots representing consecutive years of for the time period 2000 to 2100.

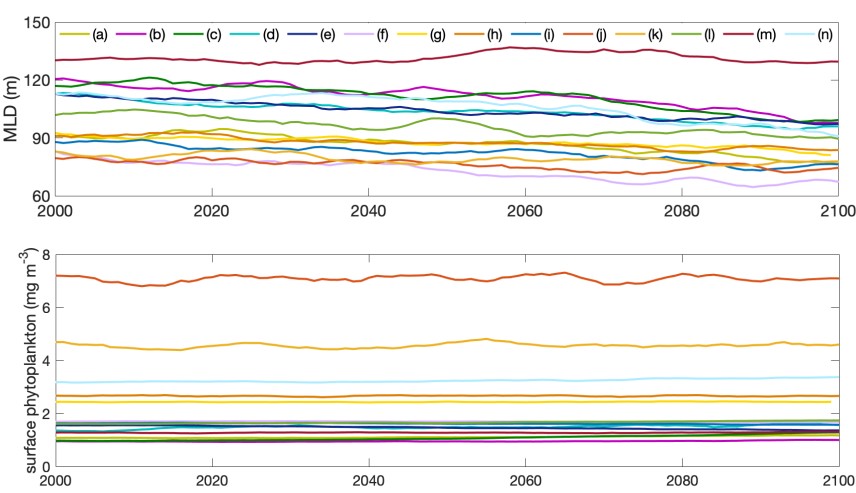

**Figure A3. Projections of mixed layer depth (top) and surface phytoplankton concentration (bottom) in absolute values from 2000 to 2100 of individual models of the multi-model ensemble in the Southern Ocean.** Different coloured lines show individual models listed in Table 1 of the main manuscript.

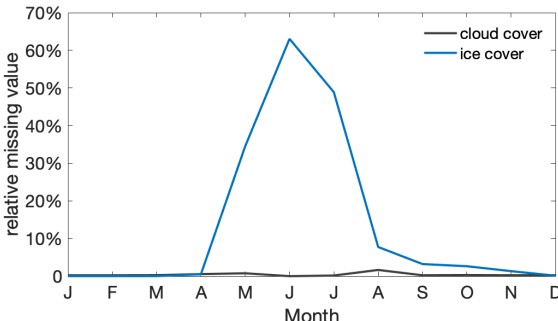

**Figure A4.** Seasonal cycle of the average portion of satellite chlorophyll missing per month and per grid box within the research area due to cloud cover (black) and ice cover/no light (blue).

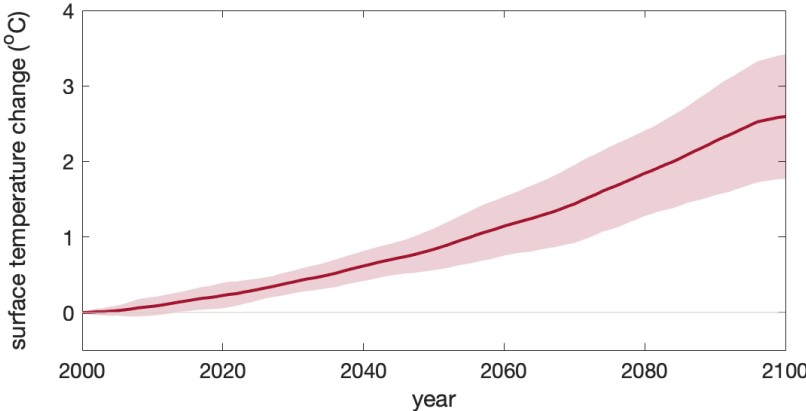

**Figure A5. Increasing sea surface temperature under climate change in the Southern Ocean.** Multi-model ensemble (MME) projections of the sea surface temperature change (unit: °C) from 2000 to 2100 under the SSP5-8.5 scenario, with shading indicating one standard deviation. The time series are filtered using a 10-year moving average.

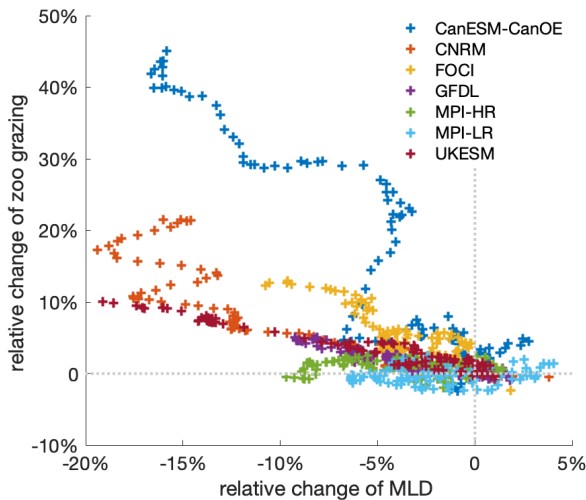

**Figure A6. Mixed layer depth directly influences zooplankton grazing by affecting phytoplankton biomass concentration, which alters prey-predator encounters.** Relation between relative anomalies of annual mean mixed layer depth and zooplankton grazing from 2000 to 2100 under the SSP5-8.5 scenario, relative to the mean values of the first decade of the 21st century (2000 - 2009) in the Southern Ocean. The time series are filtered using a 10-year moving average; different colours denote different climate models. Note, not all models include grazing as an output variable, which is why the number of models here is smaller.

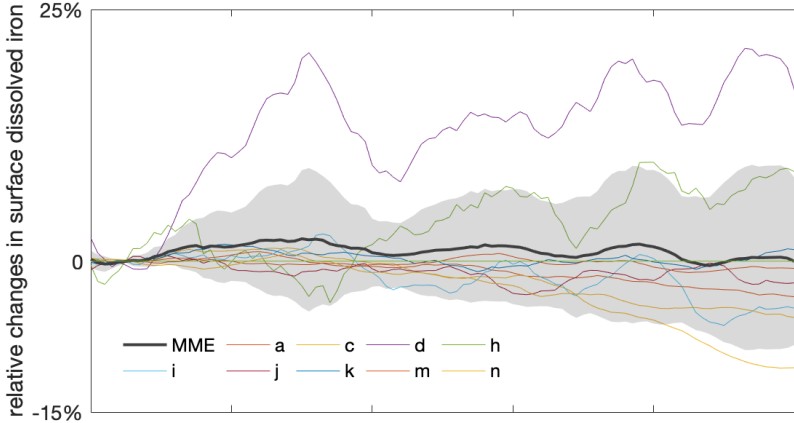

**Figure A7. Varying projected changes in future surface dissolved iron in the Southern Ocean.** Multi-model ensemble (MME) projections of the relative changes of dissolved iron from 2000 to 2100 under the SSP5-8.5 scenario, with shading indicating one standard deviation, relative to the respective mean values of the first decade of the 21 century (2000 - 2009). Dissolved iron data is only available from a limited number of models. The letter in the legend indicates the respective model listed in Table 1. The time series are filtered using a 10-year moving average.

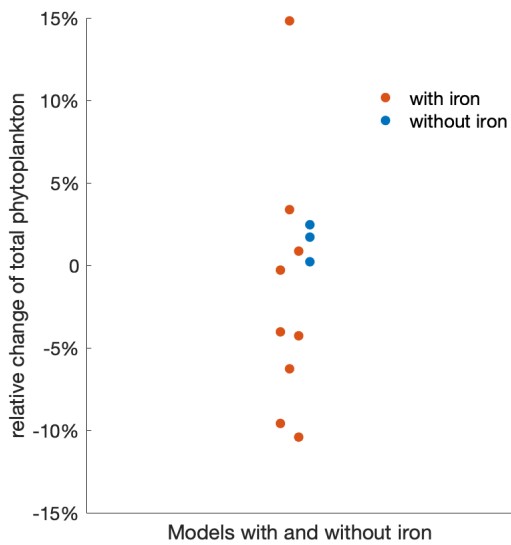

**Figure A8. ESMs that include or do not include iron limitation project the total phytoplankton change similarly, albeit with a larger spread if iron is simulated.** Beeswarm plot of the relative changes of total phytoplankton biomass over the 21st century under the SSP5-8.5 scenario using models that explicitly include iron limitation (red) and no iron limitation (blue).

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
