# Peer review of "Southern Ocean phytoplankton under climate change: shifting balance of bottom-up and top-down control"

_Biogeosciences, 2023_

## Author Comment (AC1)

Reply to reviewer 1:

*I found this manuscript interesting and generally well written. The methods are quite clear and the figures are of good quality. I appreciate their emergent constraint approach. I think this paper should be published eventually because the approach is novel and their hypothesis about phytoplankton change in the Southern Ocean is intriguing. However, there are three main points that I would like to see addressed before publication.*

Thank you for taking the time to review our manuscript. We appreciate your positive feedback and the valuable points you have raised. It is encouraging to hear that you found the manuscript interesting, well-written, and the methodology clear. We will address below the three major points you raised.

*\* I think the authors need to tone down their definitive language; as it is written now it comes off as hubris. The authors should use more speculative or conditional language to convey the points they are making. Attributing such specific mechanisms of change in phytoplankton using such a diverse set of models is problematic because the models could potentially have different processes that are controlling productivity/biomass in the Southern Ocean. Multi-model ensembles are useful in that they average out biases in individual models, but such specific attribution of mechanisms can really only be done with certainty by looking at the equations of individual models. Otherwise, it is just speculation about what's going to happen. The story that the authors describe is compelling but it does not necessarily mean that this is what's happening in every model. For example, the subantarctic region of the SO could become cloudier with climate change (see Fig 1f in Leung et al 2005), leading to an increase in Chl/C ratios of phytoplankton that could lead to increasing surface chlorophyll trend shown in Fig 6. I'm not suggesting that the authors are incorrect with their hypothesis, but they need to be more modest about how they attribute the drivers of change.*

\* Thank you for pointing this out. We have modified the manuscript to use more speculative or conditional language to better align our statements with the inherent uncertainties associated with modelling a complex system. As an example, consider the modification made in Line 150. We revised the sentence to read, "The increasing top-down control and grazing pressure on phytoplankton may be a consequence of  ...," opting for this phrasing instead of stating "is a consequence of" to tone down the certainty.

In regard to Fig 6, the effect of a change of cloud coverage on Chl:C and thereby chlorophyll is an interesting point that we will integrate into our manuscript and change Line 115 to:
 "Surface chlorophyll is used as a reliable proxy for surface phytoplankton concentration due to the strong correlation between modelled chlorophyll and phytoplankton biomass concentration (Fig. A3). This correlation holds despite the potential impact of changes in cloud coverage on the Chl:C ratio (Leung et al., 2015)."
Meanwhile, we will add Figure R1.2 below to Figure 6 to explicitly show the increase in phytoplankton biomass concentration under warming.

*\* Relating to the first point - the authors are very dismissive about the impacts of changing iron availability for phytoplankton (despite the well documented importance of iron in controlling production in the Southern Ocean; e.g., see section 3.5 and refs therein of Petrou et al., 2016). They also do not address the potentially big impact of increasing temperature.*

*As phytoplankton growth rates, zooplankton grazing rates, and phytoplankton/zooplankton loss rates are highly sensitive to temperature in most models, I think this deserves some discussion and perhaps additional analysis. How do temperature and iron conditions change in the Southern Ocean upper mixed layer in this MME? How do these changes project onto the hypothesis that the authors present?*

\* We agree that iron and temperature are two important factors influencing the phytoplankton in the research area. While we did incorporate a discussion on the impact of iron availability on phytoplankton growth and the effect of temperature on zooplankton grazing rates, we understand that our mention of these critical factors may not have been sufficiently comprehensive given their importance. In response to your suggestions, we have thoroughly revised our manuscript, enriching the existing content with a deeper exploration of the roles of temperature and iron.

To address the multifaceted impact of temperature, we have incorporated its effects into Sections 3.1 and 3.2. These sections now cover the influence of temperature on phytoplankton growth and its potential to affect zooplankton grazing. For instance, we have included the following sentence in Line 160: "Higher temperatures are linked to increased rates of zooplankton grazing, leading to more pronounced top-down control of phytoplankton populations (Caron and Hutchins, 2013), though only a few CMIP6 models include temperature-dependent grazing (Rohr et al, 2023) " We have added Fig.R1.1 bottom panel to the manuscript appendix as Figure A5, to illustrate Southern Ocean surface warming by the end of the century as simulated by the Multi-Model Ensemble (MME).

Regarding the role of iron, we have broadened our discussion to highlight the complexity of the iron cycle and the caveat of the model simulation in this aspect:
"Given the complex interplay between iron and its biological responses, along with the multitude of external processes affecting its availability, projecting changes in iron availability and subsequent phytoplankton responses under future climate conditions is challenging (Petrou et al., 2016). Current model projections do not show a significant change in iron availability (Fig.A6). As a result, models with or without iron limitation do not show distinct differences in their projections of phytoplankton responses to climate change (Fig.A7), though models with iron representation show a much larger spread than those without iron. " We have added the Fig.R1.1 top panel to the appendix—Figures A6—depicting the projected changes in dissolved iron as simulated by the MME.

[Figure]

[Figure]

Fig. R1.1 **Varying future dissolved iron trend and clearly increasing sea surface temperature under climate change**. Multi-model ensemble (MME) projections of the relative changes of dissolved iron (top) and sea surface temperature (bottom) from 2000-2100 under the SSP5-8.5 scenario, with shading indicating one standard deviation, relative to the respective mean values of the first decade of the 21 century (2000-2009) in the Southern Ocean. Dissolved iron data is only available from a limited set of models. The letter in the legend indicates the respective model listed in Table 1. The time series are filtered using a 10-year moving average.

*\* Their argument appears to be somewhat circular – The authors say that a shallower MLD leads to more concentrated phytoplankton at the surface (this is not shown) and that leads to more grazing efficiency which reduces the phytoplankton concentration (which would, in turn, reduce the grazing efficiency). So, I suggest they add more plots to show that phytoplankton biomass really is more concentrated nearer to the surface. The integrated plots that are shown in Figure 5 for example should be broken down by depth to support the hypothesis they are making. They show surface chlorophyll trends in Figure 6, but with most models having variable Chl/C ratios, this is not definitely showing what they claim. The*

*authors repeatedly say that phytoplankton concentrations in a shallower MLD increase so this needs to be demonstrated.*

\* Thank you for pointing out that we missed showing the increasing surface phytoplankton concentration in the 21st century. We did include Fig A3 in the original manuscript, showing chlorophyll and phytoplankton biomass concentration correlate very well in the model simulations. We will stress it more clearly in the method section, as mentioned above. In response to your suggestion, we will include an additional figure that explicitly illustrates the increase in phytoplankton biomass concentration over the 21st century (Fig.R1.2), supporting that it is not merely chlorophyll that increases but that phytoplankton biomass concentration does so, too.

[Figure]

Fig. R1.2 An increase in surface phytoplankton biomass concentration during the 21st century. Multi-model ensemble (MME) projections of the relative changes of surface phytoplankton biomass concentration from 2000-2100 under the SSP5-8.5 scenario, with shading indicating one standard deviation, relative to the respective mean values of the first decade of the 21st century (2000-2009) in the Southern Ocean. The time series are filtered using a 10-year moving average.

**Detailed point-by-point responses to your detailed comments are listed below:**

*L22: Rather than "poorly simulated", perhaps say "simplistic"*
R: Adapted.

*L32/33: This sentence is awkward in that the words "climate change" are used twice. Reword to something like this: " These factors are all projected to change with climate change so phytoplankton will likely be impacted from changing bottom up processes".*
R: Thanks for pointing it out. We will rephrase the sentence as follows:
"Given that these environmental factors are projected to change under climate change, it is expected that phytoplankton will also be affected."

*L40:light conditions in high latitude regions may also improve due to decreasing sea ice cover.*
R: We will add the impact of light conditions due to sea ice cover change and rephrase the sentence to:

"An opposite bottom-up response can be found in high-latitude regions where improved light conditions due to increased stratification and sea ice retreat are projected to lead to phytoplankton increases (Sarmiento et al., 2004b; Deppeler and Davidson, 2017)."

*L52: perhaps remove the word "current" since Tagliabue et al (2016) is about CMIP5 models.*
R: Thanks for pointing this out. Done.

*Figure 2 caption and throughout the Figures/captions: rather than "relative variation" could you say "relative change" or "normalized changes"… Variation implies you're looking at the variability and I find it confusing.*
R: Thanks for pointing this out. The instances of "variation" have been replaced with "relative changes".

*Figure 3 caption. Boxes denoted by dashed lines mark the focus area (reword that sentence of caption)*
R: The corresponding sentence will be replaced by the suggested sentence.

*Figure 3: are these maps means of Figures 1 and 2? If so, could you explicitly state that in the caption?*
R: Yes, indeed. We will add "Maps (a) and (b) are the means of Figures 1 and 2, respectively." for further clarification.

*L97/99: I don't think you should call out figures before you have introduced them in the Results section.*
R: We will remove the early reference to figure 9 and figure 10b.

*L150-160: this is where the logic sounds very circular – you need to show that phytoplankton concentrations actually increases in the mixed layer, or reword.*
R: Thank you for pointing out that the logic sounds circular. We will add figure R.1.2 to explicitly show that the surface phytoplankton biomass concentration projected by the MME increases by 5 ± 10% from the 2000s to the 2090s.

*L158/159: Figure 7 is not very convincing towards this point. Much of the negative relationship results from one model (CanESM).. or am I missing something?*
R: In this context, our emphasis is on the increase in zooplankton grazing as the mixed-layer depth (MLD) shoals. To support this point, the relative variation of zooplankton grazing (y-axis in Fig. 7) should exhibit an inverse relationship with the relative variation of MLD (x-axis in Fig. 7). Indeed, CanESM-CanOE demonstrates the highest sensitivity, characterized by the steepest slope. However, most other models, while displaying lower sensitivity, also clearly indicate an inverse correlation between MLD and zooplankton grazing change, namely CNRM, FOCI, GFDL and UKESM.

*L187: end this sentence after Le Quéré reference. Then start a new sentence with "It is thought to be mainly caused by…."*
R: We will rephrase the sentence as suggested to:
"Such an opposite relationship between surface chlorophyll and MLD on a seasonal scale has been previously shown in observations (Uchida et al., 2019; Arteaga et al.,2020) and

model simulations (Song et al., 2018; Arteaga et al., 2020; Le Quéré et al., 2016). It is thought to be mainly caused by seasonal dilution of phytoplankton and by growth limitation along with zooplankton grazing."

*Figure 8. I think you should be careful about using TTE in this context, as TTE depends on community composition of the plankton. For example if much of the zooplankton are microzooplankton, the energy stays more in the microbial loop, whereas if they are mesozooplankton they can be consumed by higher trophic levels. See, for example, Krumhardt et al., 2022*

R: Thank you for bringing Krumhardt et al., 2022 to our attention, which nicely shows that beyond changes in biomass, global warming is projected to influence plankton composition. A shift in plankton composition is expected to have complex repercussions on ecosystem transfer efficiency through intricate food-web interactions. Unfortunately, due to the simplicity of plankton dynamics in some models (with one phytoplankton and/or one zooplankton group) and a lack of model output (which supports our point that zooplankton-related variables should be regularly saved as standard model output), we are unable to conduct a comprehensive analysis to detect changes in plankton composition and their subsequent impact on energy transfer to higher trophic levels. At this stage, we can only define trophic transfer efficiency (TTE) in its most basic form as the ratio of depth-integrated zooplankton biomass to depth-integrated phytoplankton biomass (Eq. 5). We included this caveat in the manuscript by adding the below sentence to line 120:
"Given model simplicity and limitations of model output, the trophic transfer efficiency (TTE, Eq.5) is simply defined as the ratio of depth-integrated biomass of zooplankton to depth-integrated biomass of phytoplankton following Barnes et al., (2010)"

We will further incorporate studies about the potential change in plankton composition and its consequences through trophic interactions into our discussions (Line 228):
"Beyond changes in biomass, it has been suggested that ecosystem transfer efficiency might be affected by phytoplankton composition under global warming (Petrou et al., 2016; Krumhardt et al., 2022), an effect that we expect not to be well resolved by MME foodwebs that are simplified compared to the real ocean."

*L188-190: this is exactly why you can't definitively attribute mechanisms in the models.*
R: We will use more speculative or conditional language in the revised manuscript.

*L192-195: Have you verified that this is what is happening in each model? Otherwise please be more speculative in this statement.*
R: Thank you for pointing this out. We will tone down our language.

*L198: add 's' to 'exist'.*
R: Adapted.

*L205: rather than "mixed layer, average light" say "mixed layer with higher average light intensity".*
R: The text will be modified accordingly:
"when the MLD is shoaling and phytoplankton is being contained in a shallower surface ocean mixed layer with higher average light intensity and temperature, as a result, phytoplankton grow better and concentrations are expected to increase."

*L215: say "Our results suggest, therefore, that there will be an increase in surface chlorophyll…"*

R: We will modify the sentence accordingly:

"Our results suggest, therefore, that there will be an increase in surface chlorophyll in the subantarctic and subpolar Antarctic regions of the Southern Ocean and thus an increase in phytoplankton concentration by the end of the century."

*L218: you have not shown an increase phytoplankton concentration, so be careful how you word this statement.*

R: We will add figure R.1.2 to provide support for this statement.

*L226: add "productivity" after "phytoplankton*

R: Adapted.

*Figure 10: could you make the colors more distinguishable? The dark purple and black are so close in tone so it took me a while to see the difference in color between the two dashed lines on panel b.*

R: Thank you for pointing it out. We will make the colors more distinguishable.

*L231: replace "all" with "some"*

R: Adapted.

*L236: actually models with iron representation show a much larger spread than those without iron, which is something that should be mentioned.*

R: We will rephrase the comparison of model with and without iron to:

"Models with or without iron limitation do not reveal clear differences in the projection of phytoplankton response to climate change, though models with iron representation show a much larger spread than those without (Fig.A5)."

*L244-247: this sentence is confusing and long. Please reword. Also, mention the potential influence of different types of predator prey relationships, Holling type II and Holling type III, both of which are used in ESMs.*

R: Thank you for pointing out the long sentence. We will split it into two:

"Top-down grazing by zooplankton is related not just to the total available prey (phytoplankton biomass) but also to the efficiency with which zooplankton can feed on this prey. This efficiency is influenced by different factors including traits of both prey and predator, prey concentration, and also by the type of predator-prey relationship used in a model, such as the different Holling types."

*L255: actually, the COPEPOD dataset has pretty good coverage globally, except for the subantarctic Southern Ocean (see Moriarty and O'Brian 2013). Perhaps mention this.*

R: To acknowledge the COPEPOD dataset, we will add the below sentences after L256:

"The availability of the COPEPOD dataset (Coastal and Oceanic Plankton Ecology, Production, and Observation Database, https://www.st.nmfs.noaa.gov/copepod/, Moriarty and O'Brian 2013) has greatly enhanced access to global mesozooplankton data and provides significant support to advance relevant studies. However, the dataset does not

include microzooplankton, whose dynamics can have substantial effects on ecosystem processes, such as trophic transfer efficiency (TTE; Prowe et al. 2022 )"

*L310/311: I don't think this is true. Many studies have aimed to understand more about the zooplankton component of ESMs. See for example, Heneghan et al (2016 and 2020) & Negrete-Garcia et al (2022).*

R: The biogeochemical models we are referencing here are typically employed in climate research, specifically the Earth System Models (ESMs) included in CMIP6, as listed in Table 1. These models tend to use relatively straightforward descriptions of plankton dynamics, in contrast to more sophisticated size-spectrum models as seen in works like Heneghan et al. (2016 and 2020) and Negrete-Garcia et al. (2022), or end-to-end models capable of simulating complex trophic interactions and feedback from higher trophic levels, including fish.

We will replace "in biogeochemical models" with "in biogeochemical models for climate research, such as CMIP models."

Reference:

Leung, S., A. Cabré, and I. Marinov. "A latitudinally banded phytoplankton response to 21st century climate change in the Southern Ocean across the CMIP5 model suite." Biogeosciences 12, no. 19 (2015): 5715-5734.

Caron, D. A. and Hutchins, D. A.: The effects of changing climate on microzooplankton grazing and community structure: drivers, predictions and knowledge gaps, Journal of Plankton Research, 35, 235–252, 2013.

Rohr, T., Richardson, A. J., Lenton, A., Chamberlain, M. A., and Shadwick, E. H.: Zooplankton grazing is the largest source of uncertainty for marine carbon cycling in CMIP6 models, Communications Earth & Environment, 4, 212, 2023.

Petrou, Katherina, Sven A. Kranz, Scarlett Trimborn, Christel S. Hassler, Sonia Blanco Ameijeiras, Olivia Sackett, Peter J. Ralph, and Andrew T. Davidson. "Southern Ocean phytoplankton physiology in a changing climate." Journal of Plant Physiology 203 (2016): 135-150.

Krumhardt, Kristen M., Matthew C. Long, Zephyr T. Sylvester, and Colleen M. Petrik. "Climate drivers of Southern Ocean phytoplankton community composition and potential impacts on higher trophic levels." Frontiers in Marine Science 9 (2022): 916140.

Moriarty, R., and T. D. O'brien. "Distribution of mesozooplankton biomass in the global ocean." Earth System Science Data 5, no. 1 (2013): 45-55.

Prowe, A. F., Su, B., Nejstgaard, J. C., and Schartau, M.: Food web structure and intraguild predation affect ecosystem functioning in an established plankton model, Limnology and Oceanography, 67, 843–855, 2022.

Heneghan, Ryan F., Jason D. Everett, Julia L. Blanchard, and Anthony J. Richardson. "Zooplankton are not fish: improving zooplankton realism in size-spectrum models mediates energy transfer in food webs." Frontiers in Marine Science 3 (2016): 201.

Heneghan, Ryan F., Jason D. Everett, Patrick Sykes, Sonia D. Batten, Martin Edwards, Kunio Takahashi, Iain M. Suthers, Julia L. Blanchard, and Anthony J. Richardson. "A functional size-spectrum model of the global marine ecosystem that resolves zooplankton composition." Ecological Modelling 435 (2020): 109265.

Negrete-García, Gabriela, Jessica Y. Luo, Matthew C. Long, Keith Lindsay, Michael Levy, and Andrew D. Barton. "Plankton energy flows using a global size-structured and trait-based model." Progress in Oceanography 209 (2022): 102898.

---

## Author Comment (AC2)

Reply to reviewer 2:

*This manuscript is a nice exploration of how the plankton ecosystem in the Southern Ocean will change under a high-emissions climate change scenario. Xue et al., evaluate a set of CMIP6 models for phytoplankton, zooplankton, and mixed layer characteristics, arguing that the shift from bottom-up to top-down control under climate change results in roughly unchanged phytoplankton biomass in the Southern Ocean. Additionally, they find an emergent constraint between models' representation of the seasonal sensitivity of surface chlorophyll with the shoaling of the mixed layer to a long-term sensitivity of chlorophyll and mixed layer depth. With this emergent constraint they are able to reduce the uncertainty in model projections of chlorophyll change under climate change.*

*This is overall a nice, well-written manuscript. I found the emergent constraint portion of the manuscript to be the most interesting, however, it seemed at times disconnected from the rest of the manuscript (even though the shoaling of the ML leading to phytoplankton blooms is a key bottom-up mechanism driving phytoplankton biomass) – as if it were somehow tacked on to the manuscript at the end of the writing process rather than fully integrated from the beginning. This is evident from the fact that no mention of the emergent constraint is in the abstract or introduction, even though it makes for quite a key result.*

*While the EC result is quite interesting, I also found it to be insufficiently explored both in its own sake and also within the context of "shifting balance of bottom-up and top-down control." A few points for the authors to consider for their revision:*

We sincerely thank you for dedicating your time to reviewing our manuscript. Your feedback is highly valued, and we are genuinely appreciative of both your positive comments and the insightful points you have raised. It's encouraging to learn that you found the manuscript to be well-written and particularly found the emergent constraint aspect of the results intriguing. We consider your four main points in detail below:

*It's nice that we are able to now use this new model analysis framework of the emergent constraint to reduce uncertainty in future projections, but what does this analysis REALLY tell you about bottom-up vs. top-down control in the Southern Ocean, in terms of drivers and mechanisms? In the conclusions, you state "we further employ the approach of the emergent constraint to increase our confidence in the increasing trend of phytoplankton concentration … which is the underlying mechanism that contributes to the intensified top-down processes under climate change." (lines 297-298) but first, phytoplankton concentration (biomass) does not really increase under climate change, and second, chlorophyll as a proxy appears to be a mix of biomass and productivity. All that the EC analysis appears to really be used for is to reduce uncertainty in future projections of chlorophyll (not phytoplankton biomass). The reader is left wondering what the actual connection really is between the EC and the mechanisms of top-down vs. bottom-up control.*

R: Thank you for pointing out the disconnect between "top-down control - phytoplankton biomass concentration" and "emergent constraint - chlorophyll concentration".

Chlorophyll and phytoplankton biomass concentration correlate very well (Fig. A3). We have a figure in the appendix that supports the correlation and that we will reference in the main text not only in the methods section (as previously done) but also in the results, section 3.3.1, to more clearly link chlorophyll and biomass. Further, we will include the new Fig. R2.1 in the results section to explicitly show the change in phytoplankton biomass concentration in the 21st century. This addition will help bridge the conceptual gap and provide a more cohesive presentation of our findings.

[Figure]

Fig. R2.1 An increase in surface phytoplankton biomass concentration during the 21st century. Multi-model ensemble (MME) projections of the relative changes of surface phytoplankton biomass concentration from 2000-2100 under the SSP5-8.5 scenario, with shading indicating one standard deviation, relative to the respective mean values of the first decade of the 21st century (2000-2009) in the Southern Ocean. The time series are filtered using a 10-year moving average.

*I understand that the methodology of the EC utilizes a linear relationship between the models within the multi-model ensemble to constrain future projects. However, is there an extent to which the models (in the historical period) are sufficiently far away from the observations that they can be excluded? There are 2-3 models with a negative relationship between shoaling of the ML and chlorophyll concentration (S_seas > 0 or S_clim > 0) – what is causing this different relationship in those models? Should they be excluded, or the results interrogated further in some way?*

R: Emergent constraints are fundamentally different from model selection or weighting approaches. The models are only used to determine a relationship between a predictor variable and a predicted variable. A model that has a predictor variable far away from observations is still assumed to be equally capable of representing the relationship between that predictor variable and the predicted variable. As such, no model should be excluded and outliers in the predictor variable are even more valuable to robustly detect the underlying relationship. Only after the model-based relationship is established based on all models, this relationship is exploited with observations.

The causes of the different relationships between the seasonal variations of mixed layer and chlorophyll in individual models could potentially arise from various aspects, such as

model physics, simulation initial conditions, parameters of model equations. However, the biases in the predictor variable similarly influence the predicted variable, increasing confidence that the identified relationship is indeed robust.

In the revised manuscript, we have added the following sentences to L203 to avoid any misunderstandings:

"Though the sensitivities of chlorophyll to changes in MLD on a seasonal scale from individual models show some spread, it is important to note that the models deviating from the observed sensitivity are still considered to be capable of representing the relationship between chlorophyll sensitivity to MLD changes across seasonal and long-term scales."

*Along these veins, I noticed that many of the models clustered around the S_seas=0 line (e, i, f, n, h) all have skillful representations of mesozooplankton (Petrik et al. 2022) – and likely have put work into their modelled zooplankton such that they are not "treated stemotherly as a mere closure term." The one exception is model (c – CanESM CanOE) which is far away from the S_seas = 0 line but not particularly skillful in its representation of mesozooplankton. Have the authors thought about why this might be and what may be driving this clustering of these particular models (CMCC, UKESM, CNRM, IPSL, GFDL)?*

R: Thank you for bringing to our attention the connection to the findings in Petrik et al., 2022. In their study, Petrik et al. evaluated and used six models that simulated mesozooplankton, including CMCC, UKESM, CNRM, IPSL, GFDL, and CanESM-CanOE. Compared to the mesozooplankton observations, all models performed reasonably well, which is important for capturing the top-down process for phytoplankton. In the context of our study/the variables that we assess here (MLD, phyto seasonality), the additional eight models (include only one zooplankton group) that we include do not necessarily perform worse than the six models with multiple zooplankton groups. In fact, one of these models, the closest to observation in S_seas, k (MPI), has only one zooplankton group. To respond to your question, we include below figure R2.2 from another paper that is currently in review, using nearly identical model ensembles. This figure illustrates that biogeochemical model complexity does not systematically affect the projections on plankton change.

[Figure]

Fig. R2.2 Individual model projections of the relative changes of integrated phytoplankton (phy, horizontal axis) and zooplankton biomass (zoo, vertical axis) over the course of the 21st century, illustrating that there is no systematic effect of biogeochemical model complexity on the projections. The color of the markers indicates the number of phytoplankton groups included in the model, with light green, green, and black representing one, two, and three phytoplankton groups, respectively. The shape of the markers indicates the number of zooplankton groups included in the model, with triangle, square, and star representing one, two, and three zooplankton groups, respectively. Markers with orange edges indicate that the model explicitly includes iron limitation.

*Regarding the observational constraint – it was quite striking to me that the uncertainty around the observed chlorophyll values were so much lower than observed MLD values (Fig 9). When constructing your observed S_seas values (with uncertainties) are you comparing like to like in the MLD and surface chlorophyll fields? E.g., would it be a better comparison if you were to resample the Globcolour chlorophyll field for the 1-degree grids where Argo MLD data are available?*

R: Thank you for pointing it out. Indeed, as depicted in Figure 9, the uncertainty (standard deviation) associated with the observed MLD is notably higher compared to that of the observed chlorophyll values. We agree that this disparity is likely attributed in part to the limited availability of ARGO MLD data compared to satellite data, so that space/time are not well sampled. Additionally, another factor that might contribute to the relatively smaller uncertainty in observed chlorophyll values is the normalization process. Consider also that we here show relative changes throughout the seasonality, such normalization can effectively diminish the interannual variability of seasonality, thereby potentially reducing the uncertainty of chlorophyll to a certain extent. To this end, we will add the below sentence to state the potential source of uncertainty of the observed MLD in Line 187:

"Compared to observed chlorophyll with extended data coverage (satellite-based estimates), MLD (based on sparse in situ float data) reveals higher uncertainty, which could be largely due to the scarcity of data."

**Detailed point-by-point responses to minor comments are listed below:**

*I believe CMCC-ESM2 phytoplankton and zooplankton biomass fields are provided on the CMIP6 ESGF archive. I did a quick search today (Jan 17) and found phydiat, phymisc, zmeso, and zmicro on the archive with monthly outputs. (I did not check for ACCESS-ESM1-5).*

R: We appreciate your efforts in conducting a quick search. Indeed, we missed these models as we considered the output variable zooc that combines all zoo groups.

We will update our results and include the plankton output of CMCC in our results. We will exclude ACCESS-ESM1-5 as we find that the output for the historical simulation is missing (even though piControl and ssp585 are available).

*Figure 8a – there is no green shading to indicate the variability in phytoplankton biomass, only orange shading for the zooplankton. If you are intending for the reader to compare the orange shading in Fig. 8a with Fig. 5a then please indicate so. (Also, make your y-axis labels consistent between those two plots)*

R: thank you so much for pointing it out. We will update Figure 8a and include the green shading.

[Figure]

*It really was not clear to me what Fig. 7 was supposed to show. The text where Fig. 7 is referenced was not particularly informative – can you please expand on it (particularly for readers not familiar with Xue et al. 2022a), and if it's not essential to the main text, then perhaps remove it or place it in the supplemental?*

R: We acknowledge that Figure 7 is not clear to the readers. To further clarify the purpose of Figure 7, we will include a topic sentence (in bold) for Figure 7:

"Mixed layer depth directly influences zooplankton grazing."

Additionally, we will modified L161 to help clarify the mechanism:
"A shoaling mixed layer enhances phytoplankton concentration and supports higher prey-predator encounters for zooplankton, which in turn results in a greater grazing pressure and thus a stronger top-down control on phytoplankton."

*Again, there's no mention of the emergent constraint in the abstract or introduction. It would be great to introduce the concept of emergent constraint earlier than in the methods.*

R: Thank you for pointing this out. Indeed, we did not give emergent constraint enough credit within the abstract but will do so in the updated manuscript, Line 11:

"To increase our confidence in these projections, we employ an emergent constraint approach using the observed relationship between seasonal variations in mixed layer depth and surface chlorophyll concentration, as a proxy for surface phytoplankton concentration. This emergent constraint further supports the intensified top-down control under climate change, driven by rising phytoplankton concentrations due to shoaling mixed layers. "

*Also, the first time that chlorophyll is mentioned as a proxy for phytoplankton biomass is in section 2.4 (methods). I think that if there is space, it should be mentioned in the introduction – but also with the caveat that given variations in chl:c ratios due to photoacclimation and phytoplankton type, it is quite an imperfect proxy for phytoplankton biomass. (Though I personally think that chlorophyll is instead a proxy for a combination of phytoplankton biomass and productivity.)*

R: We acknowledge the concern regarding the varying chl:c ratio. We agree that chlorophyll is not a perfect proxy for phytoplankton biomass. However, it remains the most practical and effective option available for our study. Fig. A3 of the original shows linear relationships from different models, which indicate that chl and phytoplankton biomass are well correlated in model simulation and further support using chlorophyll as a proxy for phytoplankton biomass. In addition, we will include the new Fig.R2.1 (see response above) next to Fig. 6b (the latter shows rising chlorophyll under global warming), to explicitly show that not only chlorophyll but also phytoplankton biomass concentration is rising."

*I hope that these comments are helpful and not burdensome to address. This is indeed a nicely written paper and interesting study.*

Thank you again for your time and effort in reviewing our paper.

---

## Author Response (AR1)

Reply to reviewer #1:

Dear reviewer,

We would like to thank you for your valuable feedback and your supportive and constructive comments. Please find our point-by-point responses below:

*\* I think the authors need to tone down their definitive language; as it is written now it comes off as hubris. The authors should use more speculative or conditional language to convey the points they are making. Attributing such specific mechanisms of change in phytoplankton using such a diverse set of models is problematic because the models could potentially have different processes that are controlling productivity/biomass in the Southern Ocean. Multimodel ensembles are useful in that they average out biases in individual models, but such specific attribution of mechanisms can really only be done with certainty by looking at the equations of individual models. Otherwise, it is just speculation about what's going to happen. The story that the authors describe is compelling but it does not necessarily mean that this is what's happening in every model. For example, the subantarctic region of the SO could become cloudier with climate change (see Fig 1f in Leung et al 2005), leading to an increase in Chl/C ratios of phytoplankton that could lead to increasing surface chlorophyll trend shown in Fig 6. I'm not suggesting that the authors are incorrect with their hypothesis, but they need to be more modest about how they attribute the drivers of change.*

R: Thank you for pointing this out. We have modified the manuscript to use more speculative or conditional language to better align our statements with the inherent uncertainties associated with modelling a complex system. As an example, consider the modification made in Line 166. We revised the sentence to read, "The increasing top-down control and grazing pressure on phytoplankton may be a consequence of ...," opting for this phrasing instead of stating "is a consequence of" to tone down the certainty.

We thank the reviewer for highlighting the additional uncertainty in relating chlorophyll and phytoplankton concentrations in a future climate due to changes in cloud cover. In the revised manuscript, we have decided to constrain directly the surface phytoplankton concentration to avoid any uncertainties of different Chl/C ratios across models and potential changes in Chl/C ratios over time. In the revised manuscript, we thus updated both our emergent constraint and Figure 6 (figure 5 in revised manuscript).

*\* Relating to the first point - the authors are very dismissive about the impacts of changing iron availability for phytoplankton (despite the well documented importance of iron in controlling production in the Southern Ocean; e.g., see section 3.5 and refs therein of Petrou et al., 2016). They also do not address the potentially big impact of increasing temperature. As phytoplankton growth rates, zooplankton grazing rates, and phytoplankton/zooplankton loss rates are highly sensitive to temperature in most models, I think this deserves some discussion and perhaps additional analysis. How do temperature and iron conditions change in the Southern Ocean upper mixed layer in this MME? How do these changes project onto the hypothesis that the authors present?*

R: To address the multifaceted impact of temperature, we have incorporated some discussion of temperature on production and grazing into Sections 3.1 and 3.2. For instance, regarding the influence of temperature on phytoplankton growth, we have included in L225:
"when the MLD is shoaling and phytoplankton is being contained in a shallower surface ocean mixed layer with higher average light intensity and temperature, as a result, phytoplankton grow better and concentrations are expected to increase."

Regarding the influence of temperature on zooplankton grazing, we have included the following sentence in L174: "Additionally, higher temperatures are associated with increased zooplankton grazing rates and, thereby, a stronger top-down control on phytoplankton (Caron and Hutchins, 2013), though only a few CMIP6 models include temperature-dependent grazing (Rohr et al., 2023)."

Further, we have also added Fig.A5 to the appendix to explicitly illustrate Southern Ocean surface warming by the end of the century as simulated by the Multi-Model Ensemble (MME).

Regarding the role of iron, we now dedicate an entire paragraph in the discussion to emphasise its importance. As suggested by the reviewer, we now also added Figure A7, which explicitly illustrates how surface dissolved iron concentration is projected to change under climate change within individual models. Additionally, we provide content to highlight the complexity of the iron cycle and the caveats of model simulation in this aspect (L261-271): "Despite its importance to phytoplankton growth in the Southern Ocean, the processes of producing and cycling iron are still not yet fully understood, and it was not until the early 2000s that global ocean models began incorporating iron (Tagliabue et al., 2017; Moore et al., 2001). Even in the here used MME from the most recent generation of Earth system models, 3 out of 14 ESMs do not represent the iron cycle at all (Table 1). Of the 11 ESMs that include the iron cycle, only 9 models provide output on iron and these models simulate varying changes in iron availability in the Southern Ocean. However, across the multi-model mean, there is no discernible trend in iron concentration (Fig. A7). As a result, models with or without iron limitation do not reveal clear differences in the projection of phytoplankton responses to climate change (Fig. A8), though models with iron representation show a much larger spread than those without. Given the very different representation of the iron cycle in current ESMs and the complex interplay between iron and its biological responses, along with the multitude of external processes affecting its availability, projecting changes in iron availability likely adds a large uncertainty to phytoplankton projections under climate change (Petrou et al., 2016)."

*\* Their argument appears to be somewhat circular – The authors say that a shallower MLD leads to more concentrated phytoplankton at the surface (this is not shown) and that leads to more grazing efficiency which reduces the phytoplankton concentration (which would, in turn, reduce the grazing efficiency). So, I suggest they add more plots to show that phytoplankton biomass really is more concentrated nearer to the surface. The integrated plots that are shown in Figure 5 for example should be broken down by depth to support the hypothesis they are making. They show surface chlorophyll trends in Figure 6, but with most models having variable Chl/C ratios, this is not definitely showing what they claim. The authors repeatedly say that phytoplankton concentrations in a shallower MLD increase so this needs to be demonstrated.*

R: Thank you for pointing it out. We have updated the figure 6 (figure 5 in revised manuscript) to show explicitly the increase in surface phytoplankton concentration over the 21st century.

L22: *Rather than "poorly simulated", perhaps say "simplistic"*

R: Adapted.

L32/33: *This sentence is awkward in that the words "climate change" are used twice. Reword to something like this: " These factors are all projected to change with climate change so phytoplankton will likely be impacted from changing bottom up processes".*

R: Thanks for pointing it out. We have rephrased the sentence as follows (L38-39):
"Given that these environmental factors are projected to change under climate change, it is expected that phytoplankton will also be affected."

L40: *light conditions in high latitude regions may also improve due to decreasing sea ice cover.*

R: We have added the impact of light conditions due to sea ice cover change and rephrase the sentence to (L45-47):
"An opposite bottom-up response can be found in high-latitude regions where improved light conditions due to increased stratification and sea ice retreat are projected to lead to phytoplankton increases (Sarmiento et al., 2004b; Deppeler and Davidson, 2017)"

L52: *perhaps remove the word "current" since Tagliabue et al (2016) is about CMIP5 models.*

R: Thanks for pointing this out. Done.

*Figure 2 caption and throughout the Figures/captions: rather than "relative variation" could you say "relative change" or "normalized changes"... Variation implies you're looking at the variability and I find it confusing.*

R: Thanks for pointing this out. The instances of "variation" have been replaced with "relative changes".

*Figure 3 caption. Boxes denoted by dashed lines mark the focus area (reword that sentence of caption)*
*Figure 3: are these maps means of Figures 1 and 2? If so, could you explicitly state that in the caption?*

R: We have updated the figures by dividing Figure 3a and 3b and integrating them with Figures 1 and 2, respectively. We added a statement clarifying that the added panel represents the mean of the multi-model ensemble. Furthermore, upon suggestion, we adapted the caption to include 'Boxes denoted by dashed lines mark the focus area.

*L97/99: I don't think you should call out figures before you have introduced them in the Results section.*

R: We have removed the early reference to figure 9 and figure 10b.

*L150-160: this is where the logic sounds very circular – you need to show that phytoplankton concentrations actually increases in the mixed layer, or reword.*

R: We have updated figure 5b to explicitly show that the surface phytoplankton biomass concentration projected by the MME increases by $5 \pm 10\%$ from the 2000s to the 2090s.

*L158/159: Figure 7 is not very convincing towards this point. Much of the negative relationship results from one model (CanESM).. or am I missing something?*

R: In this context, our emphasis is on the increase in zooplankton grazing as the mixed-layer depth (MLD) shoals. To support this point, the relative variation of zooplankton grazing (y-axis in Fig. 7) should exhibit an inverse relationship with the relative variation of MLD (x-axis in Fig. 7). Indeed, CanESM-CanOE demonstrates the highest sensitivity, characterized by the steepest slope. However, most other models, while displaying lower sensitivity, also clearly indicate an inverse correlation between MLD and zooplankton grazing change, namely CNRM, FOCI, GFDL and UKESM. Given that Figure 7 in the original manuscript merely serves to underscore the impact of mixed layer depth change on top-down control—a finding consistent with existing literature—we have moved this figure to the appendix (Fig.A6).

*L187: end this sentence after Le Quéré reference. Then start a new sentence with "It is thought to be mainly caused by...."*

R: We have rephrased the sentence as suggested to:
"Such an opposite relationship between surface chlorophyll and MLD on a seasonal scale has been previously shown in observations (Uchida et al., 2019; Arteaga et al., 2020) and model simulations (Song et al., 2018; Arteaga et al., 2020; Le Quéré et al., 2016). This relationship is thought to be mainly caused by the seasonal dilution of phytoplankton and by growth limitation along with zooplankton grazing."

*Figure 8. I think you should be careful about using TTE in this context, as TTE depends on community composition of the plankton. For example if much of the zooplankton are microzooplankton, the energy stays more in the microbial loop, whereas if they are mesozooplankton they can be consumed by higher trophic levels. See, for example, Krumhardt et al., 2022*

R: At this stage, we can only define trophic transfer efficiency (TTE) in its most basic form as the ratio of depth-integrated zooplankton biomass to depth-integrated phytoplankton biomass (Eq. 5) due to simplistic plankton formation in CMIP models and lack of model output. We have incorporated a discussion on the limitations of current Earth System Models (ESMs) in capturing changes in plankton composition and have called for improvements (L252-254):

"... and the oversimplified food web formulations potentially limit the capture of phytoplankton composition changes, underscoring the need for model improvements to capture these critical changes and understand future ecosystem dynamics comprehensively (Petrou et al., 2016; Krumhardt et al., 2022)."

L188-190: *this is exactly why you can't definitively attribute mechanisms in the models.*

R: We have adapted more speculative or conditional language in the revised manuscript.

L192-195: *Have you verified that this is what is happening in each model? Otherwise please be more speculative in this statement.*

R: Thank you for pointing it out. We have toned it down.

L198: *add 's' to 'exist'.*

R: Adapted.

L205: *rather than "mixed layer, average light" say "mixed layer with higher average light intensity".*

R: The text has been modified accordingly (L225-227):
"when the MLD is shoaling and phytoplankton is being contained in a shallower surface ocean mixed layer with higher average light intensity and temperature, as a result, phytoplankton grow better and concentrations are expected to increase"

L215: *say "Our results suggest, therefore, that there will be an increase in surface chlorophyll..."*

R: We have modified the sentence in L238-239:
"Our results therefore suggest, with enhanced confidence, an increase in surface phytoplankton concentration in the subantarctic and subpolar Antarctic regions of the Southern Ocean by the end of the century (Fig. 5)."

L218: *you have not shown an increase phytoplankton concentration, so be careful how you word this statement.*

R: We have added figure 5 to provide support for this statement.

L226: *add "productivity" after "phytoplankton"*

R: Adapted.

Figure 10: *could you make the colors more distinguishable? The dark purple and black are so close in tone so it took me awhile to see the difference in color between the two dashed lines on panel b.*

R: Thank you for pointing it out. We have changed the colour from dark purple to light purple to make it more distinguishable.

L231: *replace "all" with "some"*

R: Adapted.

L236: *actually models with iron representation show a much larger spread than those without iron, which is something that should be mentioned.*

R: We have rephrased the comparison of model with and without iron to (L266-267):
"models with or without iron limitation do not reveal clear differences in the projection of phytoplankton responses to climate change (Fig. A8), though models with iron representation show a much larger spread than those without."

**L244-247:** *this sentence is confusing and long. Please reword. Also, mention the potential influence of different types of predator prey relationships, Holling type II and Holling type III, both of which are used in ESMs.*

R: Thank you for pointing out the long sentence. We have rephrased it to L287-288:
"Top-down grazing by zooplankton is influenced by factors including traits of both prey and predator, prey concentration, and also by the type of predator-prey relationship used in a model, such as the different Holling types (Kiørboe, 2009; Xue et al., 2022a; Anderson et al., 2010)."

**L255:** *actually, the COPEPOD dataset has pretty good coverage globally, except for the subantarctic Southern Ocean (see Moriarty and O'Brian 2013). Perhaps mention this.*

R: To acknowledge the COPEPOD dataset. We have added below sentences to L275:
"Despite significant advancements in ocean ecosystem monitoring over recent decades, such as the COPEPOD dataset (Coastal and Oceanic Plankton Ecology, Production, and Observation Database, https://www.st.nmfs.noaaa.gov/cope Moriarty and O'brien, 2013), supporting advanced relevant studies, a notable gap persists in observational data that specifically target higher trophic levels and data coverage in the Southern Ocean."

**L310/311:** *I don't think this is true. Many studies have aimed to understand more about the zooplankton component of ESMs. See for example, Heneghan et al (2016 and 2020) & Negrete-Garcia et al (2022).*

R: The biogeochemical models we are referencing here are typically employed in climate research, specifically the Earth System Models (ESMs) included in CMIP6, as listed in Table 1. These models tend to use relatively straight-forward descriptions of plankton dynamics, in contrast to more sophisticated size-spectrum models as seen in works like Heneghan et al. (2016 and 2020) and Negrete-Garcia et al. (2022), or end-to-end models capable of simulating complex trophic interactions and feedback from higher trophic levels, including fish.

We have replaced "in biogeochemical models" with "in biogeochemical models for climate research, such as CMIP models."

Reply to reviewer #2:

Dear reviewer,

We sincerely thank you for dedicating your time to reviewing our manuscript and for your constructive comments. Please find our point-by-point responses below:

*I found the emergent constraint portion of the manuscript to be the most interesting, however, it seemed at times disconnected from the rest of the manuscript (even though the shoaling of the ML leading to phytoplankton blooms is a key bottom-up mechanism driving phytoplankton biomass) – as if it were somehow tacked on to the manuscript at the end of the writing process rather than fully integrated from the beginning. This is evident from the fact that no mention of the emergent constraint is in the abstract or introduction, even though it makes for quite a key result. While the EC result is quite interesting, I also found it to be insufficiently explored both in its own sake and also within the context of "shifting balance of bottom-up and top-down control."*

R: Thank you for pointing that out. We have made efforts to emphasize the role of the emergent constraint more clearly and to integrate it better with the rest of the manuscript. For instance, we have modified the abstract to include the results of the emergent constraint and explain how this supports the argument for the 'increasingly important role of top-down control' from L8-16:
"A shallower mixed layer is projected on average to improve growth conditions, consequently weaken bottom-up control, and compress phytoplankton closer to the surface. The increased surface phytoplankton concentration also promotes zooplankton grazing efficiency, thus intensifying top-down control. However, large differences across the model ensemble exist, with some models simulating a decrease in surface phytoplankton concentrations. To reduce uncertainties of surface phytoplankton concentration projections, we employ an emergent constraint approach using the observed sensitivity of surface chlorophyll concentration, used as an observable proxy for phytoplankton, to seasonal changes in the mixed layer depth as an indicator for future changes in surface phytoplankton concentrations. The emergent constraint reduces uncertainties of phytoplankton concentration projections by around one third and increases confidence that phytoplankton concentrations will indeed rise due to shoaling mixed layers under global warming, thus favouring intensified top-down control."

*It's nice that we are able to now use this new model analysis framework of the emergent constraint to reduce uncertainty in future projections, but what does this analysis REALLY tell you about bottom-up vs. top-down control in the Southern Ocean, in terms of drivers and mechanisms? In the conclusions, you state "we further employ the approach of the emergent constraint to increase our confidence in the increasing trend of phytoplankton concentration ... which is the underlying mechanism that contributes to the intensified top-down processes under climate change." (lines 297-298) but first, phytoplankton concentration (biomass) does not really increase under climate change, and second, chlorophyll as a proxy appears to be a mix of biomass and productivity. All that the EC analysis appears to really be used for is to reduce uncertainty in future projections of chlorophyll (not phytoplankton biomass). The reader is left wondering what the actual connection really is between the EC and the mechanisms of top-down vs. bottom-up control.*

R: Thank you for highlighting the disconnect between 'top-down control - phytoplankton biomass concentration' and 'emergent constraint - chlorophyll concentration.' To streamline the narrative and make the connection clearer for the reader, we have modified Figure 5 (originally Figure 6) to replace the projection of surface chlorophyll with that of surface phytoplankton concentration directly. This figure explicitly shows that phytoplankton concentration is projected to increase by $5 \pm 10\%$ by the end of the century. Furthermore, based on the strong correlation between projected chlorophyll and phytoplankton biomass concentration (Fig. A2), we have also adapted the emergent constraint to focus directly on the long-term sensitivity of surface phytoplankton concentration to MLD change, rather than the previously presented sensitivity of surface chlorophyll concentration to MLD change (Fig. 8; an example usage of emergent relationship between two different variables can be seen in Terhaar et al., 2021). After adjusting the focus to long-term sensitivity, the emergent constraint still demonstrates a correlation of modeled seasonal and long-term sensitivity with $R^2=0.65$ (previously $R^2=0.56$), and it successfully reduces the uncertainty of surface phytoplankton concentration by 34%. (Fig. 5).

*I understand that the methodology of the EC utilizes a linear relationship between the models within the multi-model ensemble to constrain future projects. However, is there an extent to which the models (in the historical period) are sufficiently far away from the observations that they can be excluded? There are 2-3 models with a negative relationship between shoaling of the ML and chlorophyll concentration ($S_{seas} > 0$ or $S_{clim} > 0$) – what is causing this different relationship in those models? Should they be excluded, or the results interrogated further in some way?*

R: Emergent constraints are fundamentally different from model selection or weighting approaches. The models are only used to determine a relationship between a predictor variable and a predicted variable. A model that has a predictor variable far away from observations is still assumed to be equally capable of representing the relationship between that predictor variable and the predicted variable. As such, no model should be excluded and outliers in the predictor variable are even more valuable to robustly detect the underlying relationship. Only after the model-based relationship is established based on all models, this relationship is exploited with observations. The causes of the different relationships between the seasonal variations of mixed layer and chlorophyll in individual models could potentially arise from various aspects, such as model physics, simulation initial conditions, parameters of model equations. However, the biases in the predictor variable similarly influence the predicted variable, increasing confidence that the identified relationship is indeed robust. In the revised manuscript, we have added the following sentences to L221 to avoid any misunderstandings:
"Though the sensitivities of chlorophyll to changes in MLD on a seasonal scale from individual models show some spread, it is important to note that the models deviating from the observed sensitivity are still considered to be capable of representing the relationship between chlorophyll sensitivity to MLD changes across seasonal and long-term scales."

*Along these veins, I noticed that many of the models clustered around the $S_{seas}=0$ line (e, i, f, n, h) all have skillful representations of mesozooplankton (Petrik et al. 2022) – and likely have put work into their modelled zooplankton such that they are not "treated stemotherly as a mere closure term." The one exception is model (c – CanESM CanOE) which is far away from the $S_{seas} = 0$ line but not particularly skillful in its representation of mesozooplankton. Have the authors thought about why this might be and what may be driving this clustering of these particular models (CMCC, UKESM, CNRM, IPSL, GFDL)?*

R: Thank you for bringing to our attention the connection to the findings in Petrik et al., 2022. In their study, Petrik et al. evaluated and used six models that simulated mesozooplankton, including CMCC, UKESM, CNRM, IPSL, GFDL, and CanESM-CanOE. Compared to the mesozooplankton observations, all models performed reasonably well, which is important for capturing the top-down process for phytoplankton. In the context of our study, the variables that we assess here (MLD, phyto seasonality), the additional eight models (include only one zooplankton group) that we include do not necessarily perform worse than the six models with multiple zooplankton groups. In fact, one of these models, the closest to observation in $S_{seas}$, k (MPI), has only one zooplankton group. What we found in another study is that biogeochemical model complexity (number of phytoplankton and zooplankton groups) does not systematically affect the projections on plankton change.

*Regarding the observational constraint – it was quite striking to me that the uncertainty around the observed chlorophyll values were so much lower than observed MLD values (Fig 9). When constructing your observed $S_{seas}$ values (with uncertainties) are you comparing like to like in the MLD and surface chlorophyll fields? E.g., would it be a better comparison if you were to resample the Globcolour chlorophyll field for the 1-degree grids where Argo MLD data are available?*

R: Thank you for pointing it out. Indeed, as depicted in Figure 9, the uncertainty (standard deviation) associated with the observed MLD is notably higher compared to that of the observed chlorophyll values. We agree that this disparity is likely attributed in part to the limited availability of ARGO MLD data compared to satellite data, so that space and time are not well sampled. Additionally, another factor that might contribute to the relatively smaller uncertainty in observed chlorophyll values is the normalization process. Consider also that we here show relative changes throughout the seasonality, such normalization can effectively diminish the interannual variability of seasonality, thereby potentially reducing the uncertainty of chlorophyll to a certain extent. To this end, we will add the below sentence to state the potential source of uncertainty of the observed MLD in L202-203:

" MLD reveals higher uncertainty compared to the extensively covered observed chlorophyll (satellite-based estimates), likely due to the scarcity of in situ-based data used to estimate the MLD."

*I believe CMCC-ESM2 phytoplankton and zooplankton biomass fields are provided on the CMIP6 ESGF archive. I did a quick search today (Jan 17) and found phydiat, phymisc, zmeso, and zmicro on the archive with monthly outputs. (I did not check for ACCESS-ESM1-5).*

R: We appreciate your efforts in conducting a quick search. Indeed, we overlooked these models because we considered the output variable 'zooc,' which combines all zooplankton groups. Furthermore, we found that, despite lacking vertical information, the ACCESS-ESM1-5 model provides surface phytoplankton concentration data (phycos). Therefore, in the revised manuscript, we have updated our results to include the plankton output from CMCC. Additionally, we have incorporated the ACCESS model's surface phytoplankton concentration data into the emergent constraint.

*Figure 8a – there is no green shading to indicate the variability in phytoplankton biomass, only orange shading for the zooplankton. If you are intending for the reader to compare the orange shading in Fig. 8a with Fig. 5a then please indicate so. (Also, make your y-axis labels consistent between those two plots)*

R: Thank you so much for pointing it out. We have updated now figure 6a and include the green shading.

*It really was not clear to me what Fig. 7 was supposed to show. The text where Fig. 7 is referenced was not particularly informative – can you please expand on it (particularly for readers not familiar with Xue et al. 2022a), and if it's not essential to the main text, then perhaps remove it or place it in the supplemental?*

R: We acknowledge that Figure 7 is not clear to the readers. Given that Figure 7 in the original manuscript merely serves to underscore the impact of mixed layer depth change on top-down control—a finding consistent with existing literature—we have moved this figure to the appendix (Fig.A6).

*Again, there's no mention of the emergent constraint in the abstract or introduction. It would be great to introduce the concept of emergent constraint earlier than in the methods.*

R: Thank you for pointing this out. Indeed, we did not give emergent constraints enough credit within the abstract. We have adapted the abstract according to the suggestions as shown above.

*Also, the first time that chlorophyll is mentioned as a proxy for phytoplankton biomass is in section 2.4 (methods). I think that if there is space, it should be mentioned in the introduction – but also with the caveat that given variations in chl:c ratios due to photoacclimation and phytoplankton type, it is quite an imperfect proxy for phytoplankton biomass. (Though I personally think that chlorophyll is instead a proxy for a combination of phytoplankton biomass and productivity.)*

R: We acknowledge the concerns regarding the varying Chl:C ratio. To further substantiate the correlation between chlorophyll and phytoplankton biomass, we included Figure A2. This figure demonstrates linear relationships from different models, indicating that chlorophyll and phytoplankton biomass are well correlated in model simulations. Therefore, in the revised manuscript, we have decided to constrain directly the surface phytoplankton concentration to avoid any uncertainties of different Chl/C ratios across models and potential changes in Chl/C ratios over time.

---

## Author Response (AR2)

Reply to reviewer #1:

Dear reviewer,

We would like to thank you for the time you spent on the second review. Please find our point-by-point responses below:

*\* I think it needs to better emphasized that \*surface\* phytoplankton concentration increases, while total (depth-integrated) phytoplankton concentrations remain constant or slightly decrease. This is critical for understanding Figure 5b and Figure6a. Also, the title of Section 3.2 should include the word "surface", otherwise it's contradicting what was just shown in Figure 6a. Same for the section titles of 3.3 & 3.3.1. Also please add "surface" around line 15 of the Abstract in the revised manuscript.*

R: Thank you for pointing this out. We have modified the manuscript accordingly. Specifically, we added the word "surface" in the abstract (lines 14 and 15), as well as in the method section (lines 99, 105, and 132). Furthermore, we updated the titles of Section 3.2 and Section 3.3 & 3.3.1. Additionally, the word "surface" has been included in lines 239, 241, 243, 248, 251, 258, 339, and 344.

Reply to reviewer #2:

Dear reviewer,

We appreciate your second review and your supportive, constructive feedback. Below, please find our point-by-point responses:

*I commend the authors on a fairly thorough addressing of the reviewer comments. Overall, I found their revisions satisfactory. However, I remain confused on the modifications that they made to their emergent constraint analyses. It is not clear to me how they justified using an observable contemporary sensitivity between chlorophyll and MLD to then constrain future projections between phytoplankton biomass and MLD. In the response to reviewers they just cite Terhaar et al. 2021, but I think it would benefit the readers of this manuscript to have a more thorough justification in the text. A positive correlation with chlorophyll and phytoplankton biomass is implied here, but how strong of a relationship do you need to have in order to justify such an approach?*

R: Thank you for highlighting the need for additional clarification regarding the utilisation of different variables for our emergent constraints. Accordingly, we have included further elucidation on past instances where emergent constraints were employed with both identical and diverse variables, as added in L90-96.

"Emergent constraints are often built upon relationships between the same variable at different times, e.g., Kwiatkowski et al. [2017] establish a link between the change in tropical primary production in response to temperature changes on interannual timescales and the change in tropical primary production in response to temperature changes over the 21$^{st}$ century. However, emergent constraints can also be built on relationships between different variables if these are mechanistically related, e.g., Terhaar et al. [2021a] used Southern Ocean sea surface salinity to constrain future uptake of anthropogenic carbon in that region because sea surface salinity determines sea surface density and hence the amount of mode and intermediate water formation."

Additionally, we have provided a more detailed explanation of why we chose different variables for our emergent constraint, as described now in L101-L104:

"The emergent constraint we utilised here, incorporating distinct observable and constraint variables, hinges on the relationship between surface chlorophyll concentration and surface phytoplankton biomass concentration across individual models (Fig. A2). Benefiting from its comprehensive spatial-temporal coverage, chlorophyll provides an ideal balance by offering a strong linear correlation, easy observational access, and comparatively low observational uncertainties."